# Water-assisted hydrogen spillover in Pt nanoparticle-based metal–organic framework composites

Zhida Gu [1,2], Mengke Li[2], Cheng Chen[2], Xinglong Zhang[2], Chengyang Luo[2], Yutao Yin[2], Ruifa Su[2], Suoying Zhang[2], Yu Shen[3], Yu Fu [1] ✉, Weina Zhang [2] ✉ & Fengwei Huo [2] ✉

Hydrogen spillover is the migration of activated hydrogen atoms from a metal particle onto the surface of catalyst support, which has made significant progress in heterogeneous catalysis. The phenomenon has been well researched on oxide supports, yet its occurrence, detection method and mechanism on non-oxide supports such as metal–organic frameworks (MOFs) remain controversial. Herein, we develop a facile strategy for efficiency enhancement of hydrogen spillover on various MOFs with the aid of water molecules. By encapsulating platinum (Pt) nanoparticles in MOF-801 for activating hydrogen and hydrogenation of C=C in the MOF ligand as activated hydrogen detector, a research platform is built with Pt@MOF-801 to measure the hydrogenation region for quantifying the efficiency and spatial extent of hydrogen spillover. A water-assisted hydrogen spillover path is found with lower migration energy barrier than the traditional spillover path via ligand. The synergy of the two paths explains a significant boost of hydrogen spillover in MOF-801 from imperceptible existence to spanning at least 100-nm-diameter region. Moreover, such strategy shows universality in different MOF and covalent organic framework materials for efficiency promotion of hydrogen spillover and improvement of catalytic activity and antitoxicity, opening up new horizons for catalyst design in porous crystalline materials.

Since its first discovery in 1964[1], hydrogen spillover has been extensively researched and brought about significant progress in the fields of hydrogen storage[2,3], sensing[4], and especially heterogeneous catalysis[5–7]. Hydrogen spillover refers to $H_2$ cleavage on an active metal followed by the migration of dissociated H atoms to the support[8–11], which is considered as an important step in different hydrogenation or hydrogenolysis reactions[12–14]. This effect not only explains the synergistic relationship between metals and different types of supports, but also makes it possible to control the catalytic sequences separately and

precisely towards more efficient synthesis of chemicals[15–17]. To date, studies on hydrogen spillover have mainly focused on oxide supports such as reducible metal oxide supports (i.e., iron oxide[18] and titanium oxide[19]) and non-reducible metal oxide supports (such as $Al_2O_3$[20]). In the former case, the migration mechanism of activated hydrogen atoms is generally concluded to be coherent proton-electron movement, in which the H atom donates its electron to a reducible metal cation of the oxide support and diffuses in the form of proton[21]. As for the latter, the existence of hydrogen spillover on a non-reducible oxide

[1]College of Science, Northeastern University, Shenyang 100819, China. [2]Key Laboratory of Flexible Electronics (KLOFE) & Institute of Advanced Materials (IAM), School of Flexible Electronics (Future Technologies), Nanjing Tech University (NanjingTech), Nanjing 211816, China. [3]State Key Laboratory of Organic Electronics and Information Displays (SKLOEID), Institute of Advanced Materials (IAM), Nanjing University of Posts & Telecommunications, Nanjing 210023, China. ✉e-mail: fuyu@mail.neu.edu.cn; iamwnzhang@njtech.edu.cn; iamfwhuo@njtech.edu.cn

has been controversial because of previous undesirable theoretical predictions. Although its existence has been observed on $Al_2O_3$ recently[22], the hydrogen spillover is restricted in short distance (less than sub-nanometer), and its arguable mechanism (metal migration, surface defect, or surface contamination) needs to be further investigated. Furthermore, the perplexity of mechanism and restricted spillover distance exist not only on the non-reducible metal oxide supports, but also on the non-oxide supports such as metal–organic frameworks (MOFs), which are a kind of popular catalytic support in heterogeneous catalysis[23]. Gaining insight into the mechanisms will favor the migration efficiency of activated hydrogen atoms, which can be used to design ideal MOF catalysts.

MOFs are a type of porous crystalline material with highly ordered periodic networks, which are coordinated by metal ions or clusters with organic ligands[24]. Due to their intriguing properties such as large surface area[25], uniform channel[26], tailorable chemistry[27], adjustable morphology[28] and designable component[29], MOFs exhibit unique potential in many fields, especially catalysis[30−32]. However, due to the lack of detection methods, the existence of hydrogen spillover in MOFs was in suspense for a long time. Via direct metal doping in MOFs, the capacity promotion of hydrogen storage was once considered as a proof of the existence of hydrogen spillover[33]. Further studies found that as there is no hydrogen exchange between $H_2$ and the framework's hydrogen atoms in MIL-101(Cr)[34] and HKUST-1[35], the enhanced Pd–H on the Pd surface contributes to the hydrogen storage capacity promotion rather than hydrogen spillover. Recently, Zeng's group designed multishell nano-matryoshka-structured ZIFs, in which ZIF-8 layer served as a ruler to directly evidence the hydrogen spillover in ZIF-8 by the decomposition of ZIF-67 as detector[23]. Although these reports provide a valuable exploration of hydrogen spillover in MOFs, this research field is still in its infancy and faced with significant challenges, several of which we attempt to address here. Firstly, the slow hydrogen atom mobility rate and short spillover distance in MOFs increase the difficulty in the detection and application of hydrogen spillover. Secondly, the poor thermal and chemical stability of most MOFs compromise the spillover promotion efficiency of most reported strategies including temperature regulation[36] and chemical modification[37]. Thirdly, compared with hydrogen spillover investigations on oxide supports, the

complicated MOF microenvironments including large surface area, abundant pore structure, organic–inorganic hybrid component, and coordinated guest molecules pose challenges to the characterization and mechanism research of hydrogen spillover in MOFs.

However, from another point of view, intricate MOF microenvironments may also bring about more opportunities for hydrogen spillover, such as increased adsorption and reaction sites by introducing various kinds of guest molecules into MOF pores or matrixes[38]. In fact, many chemicals, especially water molecules, are found to remarkably promote the efficiency of hydrogen spillover on various supports including $FeO$[39], $Fe_3O_4$[40], $Pt/Ba/Al_2O_3$[41], and graphene[42] by forming and stabilizing OH groups on the support surface[37] or accommodating activated hydrogen atoms as a medium. As it is well known, many MOFs are recognized for their water absorption capacity on their clusters or ligands[43,44], which may provide more possibilities for hydrogen spillover in efficiency promotion and mechanism exploration. Herein, a facile strategy was developed to enhance the efficiency of hydrogen spillover in MOFs via water molecules as medium. A research platform was accordingly designed for evaluating hydrogen spillover in MOFs with ligand hydrogenation as a detector. This platform can not only accurately quantify the efficiency and distance of hydrogen spillover but also demonstrate the migration path of activated hydrogen atoms in MOFs. Specifically, Pt@MOF-801 was designed by ingeniously encapsulating Pt nanoparticles (Pt NPs) in the center of MOF-801, in which Pt NPs served as hydrogen dissociation sites while the C=C in the MOF ligand as the acceptor of activated hydrogen atoms (Fig. 1). By thermal treatment with $H_2$ at 200 °C, the efficiency of hydrogen spillover could be evaluated by ligand conversion and MOF structural changes in MOF-801. Without water assistance, imperceptible ligand conversion suggested negligible hydrogen spillover in MOF-801 due to the high hydrogen migration energy barrier of the C–C single bond[45]. In contrast, the introduction of water molecules brought about a great enhancement of hydrogen spillover in MOF-801 spanning a region of around 100 nm in diameter at atmospheric pressure and even a region of more than 150 nm in diameter under higher humidity and pressure. Such efficiency enhancement of hydrogen spillover is attributed to the construction of a water-assisted spillover path with lower hydrogen migration energy barriers than

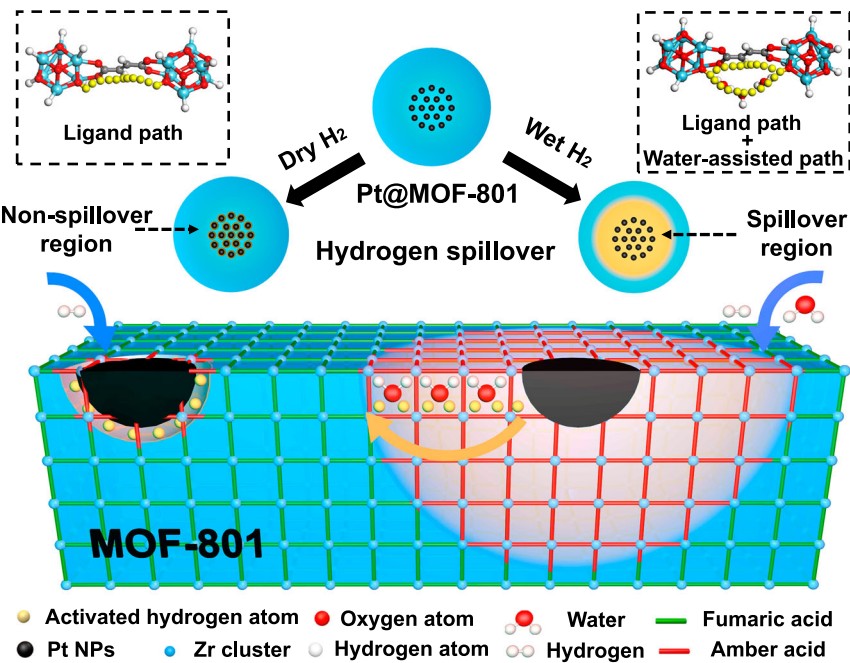

**Fig. 1 | Scheme of hydrogen spillover in MOF-801 under dry or wet $H_2$ conditions.** The activated hydrogen atom (yellow ball) by dissociation from Pt nanoparticles (black ball) in Pt@MOF-801 hydrogenates the MOF-801 ligand from fumaric acid (green stick) to amber acid (red stick) as hydrogen spillover detector.

those in the traditional ligand spillover path, as calculated by density function theory (DFT). This water-assisted spillover path, which offers dynamic advantages in the diffusion of activated hydrogen atoms, is formed with water chain between zirconium-oxide clusters[46] in the confined MOF nanochannel. Moreover, this strategy is not limited to MOF-801, but is also extendable to more MOFs or covalent organic frameworks (COFs) with a similar promotion phenomenon of hydrogen spillover. Hence, porous materials in catalysis are expected to become active centers for hydrogenation rather than only substrate materials as used in the past, which may open up new horizons for designing novel catalysts with superior chemo-selectivity and activity.

## Results

### Structure and stability of MOF-801

MOF-801 is a type of MOF with an *fcu* topology (Supplementary Fig. 1). It consists of $Zr_6O_4(OH)_4(-CO_2)_n$ as clusters and fumaric acid as the ligands, which was first reported by Peter Behrens et al.[47]. Due to its similar topological structure to UiO-66, MOF-801 demonstrates excellent thermal stability (260 °C in air), superior water absorption capacity at low humidity (2.8 liters of water per kilogram of MOF per day at 20% relative humidity (RH))[48], and water stability for at least a week[47]. In our strategy, MOF-801 was selected as the research model by ligand conversion to intuitively demonstrate hydrogen spillover in MOFs with at least four advantages. Firstly, hydrogenation is easier to characterize than the signal of hydrogen spillover. Secondly, the hydrogenation of the C=C bond can directly evidence H diffusion on the ligand, which is beneficial for path identification of hydrogen spillover in MOFs. Thirdly, the research on the water absorption capacity and mechanism of MOF-801 is relatively clear and adequate, providing a theoretical basis and model for investigating water-assisted hydrogen spillover. Lastly, compared with other signal conversions (such as tungsten oxide discoloration, oxide reduction, or hydrogenolysis), hydrogenation can better demonstrate the hydrogenation ability and potential application of hydrogen spillover for catalysis. Here, MOF-801 and Pt@MOF-801 were synthesized by the reported literature[49] and the controlled nanoparticle encapsulation strategy[50], respectively. Specifically, MOF-801 (Supplementary Figs. 2a, 3a) and Pt@MOF-801 (Supplementary Fig. 2b, c) had uniform spherical morphologies with around 180, 150, and 110 nm diameter, respectively, as revealed in the transmission electron microscopy (TEM) images (Supplementary Fig. 4). In addition, Pt NPs with a size of 4 nm (Supplementary Fig. 3d) were encapsulated in the central range (50 nm diameter) of both Pt@MOF-801 samples, which was beneficial to creating a hydrogen spillover region (Supplementary Fig. 3b, c). Powder X-ray diffraction (PXRD) patterns (Supplementary Fig. 5) showed perfect peaks matching between these samples and the MOF-801 simulation. The Brunauer–Emmett–Teller (BET) surface areas of MOF-801 and the two Pt@MOF-801 were 986, 938, and 937 $m^2 \cdot g^{-1}$, respectively. In addition, all of them had similar nitrogen adsorption–desorption isotherms (Supplementary Fig. 6) and pore size distributions (Supplementary Fig. 7) at 0.6 and 1.2 nm, which were in accordance with the reported results in the literature[51]. These data demonstrated that the incorporation of Pt NPs has negligible impact on the crystalline structure and pore size distribution of MOF-801. From the thermogravimetric analysis (TGA) (Supplementary Fig. 8), MOF-801 and Pt@150 nm MOF-801 exhibited a decline at 50–100 °C for loss of free water molecules and became constant at 100–210 °C, which confirmed that the thermostability of MOF-801 can reach 200 °C as the literature reported[52]. In addition, MOF-801 exhibited outstanding cycling performance and showed reliable stability in the water uptake for all five cycles[53]. Furthermore, more control experiments on the stability of MOF-801 were investigated including the effects of $H_2$, humidity and Pt NPs. Specifically, MOF-801 in wet $H_2$ (28 °C and 85% RH) thermal treatment at 200 °C (Supplementary Fig. 9a) and Pt@150 nm MOF-801 in wet $N_2$ (28 °C and 80% RH) thermal

treatment at 200 °C (Supplementary Fig. 9b) were tested, respectively. Both of them presented similar diffraction peaks (Supplementary Fig. 10), indicating the preservation of crystalline structure. Meanwhile, similar nitrogen adsorption–desorption curves (Supplementary Fig. 11) and pore size distributions (Supplementary Fig. 12) were also displayed and confirmed the pore structure. These results evidenced that 200 °C is an optimal temperature for hydrogen spillover exploration in MOF-801.

### Experimental evidences of water-assisted hydrogen spillover in MOF-801

To investigate efficiency enhancement of hydrogen spillover by assistance of water molecules in MOF-801, samples including Pt@150 nm MOF-801, Pt@110 nm MOF-801 and mixtures of Pt NPs with MOF-801 (named as Pt-MOF-801) were, respectively, treated at 200 °C for 2 h in input $H_2$ under two humidity conditions: dry $H_2$ (28 °C and 20% RH) and wet $H_2$ (28 °C and 85% RH). In addition, before the heat treatment in $H_2$, a thermal pretreatment in $N_2$ with the same temperature and humidity for 2 h was performed to eliminate the distractions of the original moisture in MOF (Supplementary Fig. 13). Here, samples including Pt@150 nm MOF-801, Pt@110 nm MOF-801, Pt-MOF-801, and other Pt@MOFs or Pt/COFs were named as sample(dry) after dry $N_2$ and $H_2$ thermal treatment for respective 2 h at 200 °C, and sample(wet) after wet $N_2$ and $H_2$ thermal treatment for respective 2 h at 200 °C. Pt@150 nm MOF-801(dry), Pt@110 nm MOF-801(dry) and Pt-MOF-801(dry) preserved their pristine morphology after the dry $H_2$ thermal treatment (Fig. 2a–c, and Supplementary Fig. 14a–c). The Pt@150 nm MOF-801(wet) (Fig. 2d and Supplementary Figs. 14e, 15a) was found to maintain the morphology, while the Pt@110 nm MOF-801(wet) (Fig. 2e and Supplementary Figs. 14f, 15b) exhibited additional adhesion phenomenon between MOF particles. During the thermal treatment in wet $H_2$ at 200 °C, amber acid which was generated by hydrogenation of fumaric acid ligand would melt probably due to the lower melting point (188 °C). The melted amber acid might be the reason behind this adhesion on the surface of MOF nanoparticles, which could be supported by the absence of adhesion in wet $H_2$ at 160 °C (Supplementary Fig. 16). Moreover, this inference was confirmed by more obvious adhesion between Pt-MOF-801(wet) particles (Fig. 2f and Supplementary Figs. 14d, 15c) due to the shorter spillover distance than that in Pt@110 nm MOF-801(wet) and the higher humidity than that in Pt-MOF-801(dry) (Fig. 2f). Therefore, the hydrogen spillover in wet $H_2$ may reach exactly the surface of Pt@110 nm MOF-801(wet) but not the surface of Pt@150 nm MOF-801(wet). After the dry $H_2$ treatment, the samples retained all the characteristic peaks of MOF-801 (Fig. 2g). On the contrary, after the wet $H_2$ treatment, the samples exhibited broaden diffractions at 8° and 30° in the PXRD pattern, indicating the amorphization of MOF-801 from ligand hydrogenation. According to the nitrogen adsorption–desorption measurements, the BET surface areas of Pt@150 nm MOF-801(dry), Pt@110 nm MOF-801(dry), and Pt-MOF-801(dry) were 923, 873, and 919 $m^2 \cdot g^{-1}$, which were quite similar to the BET surface area before the dry $H_2$ treatment (Fig. 2i). Meanwhile, the BET surface areas of Pt@150 nm MOF-801(wet), Pt@110 nm MOF-801(wet) and Pt-MOF-801(wet) were 495, 266, and 539 $m^2 \cdot g^{-1}$, which were significantly reduced compared with those before the wet $H_2$ treatment. Moreover, these samples had similar pore size distributions (Fig. 2j) to MOF-801 after the dry/wet $H_2$ treatments, but were decreased in pore volume after the wet $H_2$ treatment, suggesting partial porous structure distortion in MOF-801. The hydrogenation of fumaric acid ligand would lead to the loss of the planar rigid structure of C=C bonds, which induced the structural destruction in MOF-801. Typically, during thermal treatment, the ligand hydrogenation by hydrogen spillover in MOF-801 could be reflected by the $H_2$ consumption. Thus, Pt@150 nm MOF-801, Pt@110 nm MOF-801, Pt-MOF-801, Pt@150 nm MOF-801(wet), Pt@110 nm MOF-801(wet), Pt-

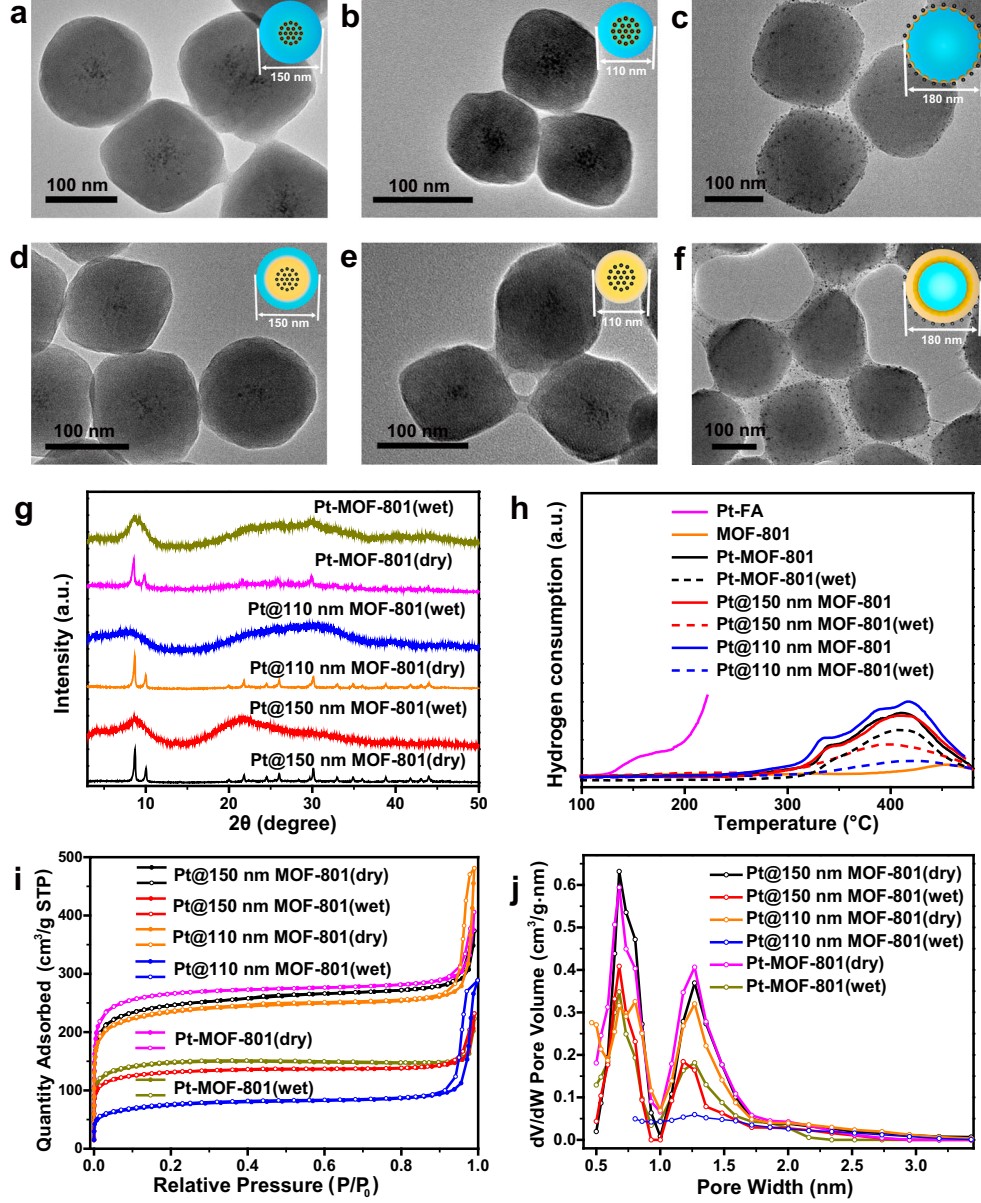

**Fig. 2 | Structure characterization of Pt@110 nm MOF-801, Pt@150 nm MOF-801, and Pt-MOF-801 after dry or wet H₂ thermal treatments.** TEM images of **a** Pt@150 nm MOF-801(dry), **b** Pt@110 nm MOF-801(dry), **c** Pt-MOF-801(dry), **d** Pt@150 nm MOF-801(wet), **e** Pt@110 nm MOF-801(wet) and **f** Pt-MOF-801(wet). **g** PXRD patterns, **h** TPR analysis, **i** nitrogen adsorption–desorption measurements, and **j** pore size distribution of Pt@150 nm MOF-801(dry), Pt@110 nm MOF-801(dry), Pt-MOF-801(dry), Pt@150 nm MOF-801(wet), Pt@110 nm MOF-801(wet), and Pt-MOF-801(wet).

MOF-801(wet), and the mixtures of Pt NPs with fumaric acid (named as Pt-FA) were, respectively, examined by temperature-programmed reduction (TPR) in $H_2$ (Fig. 2h). Pt-FA showed a very obvious hydrogen consumption from 120 °C, which implied that fumaric acid can be hydrogenated over 120 °C. All other samples showed hydrogen consumption from 300 °C, implying the occurrence of hydrogen spillover in MOF-801 above 300 °C. However, 300 °C was too high for MOF-801 to preserve its structure and resulted in serious structural destruction, leading to negligible hydrogen spillover in MOF-801 (Supplementary Figs. 17–20). The Pt@150 nm MOF-801(wet), Pt@110 nm MOF-801(wet) and Pt-MOF-801(wet) initially showed similar tendencies at 300 °C but obvious lower intensities of hydrogen consumption at 400 °C than themselves without wet $H_2$ treatment. This intensity difference in hydrogen consumption can be attributed to hydrogen pre-consumption during the wet $H_2$ treatment of samples due to hydrogen spillover. The above characterizations all proved a significant

improvement of ligand hydrogenation in wet $H_2$ compared to that in dry $H_2$, implying hydrogen spillover enhancement. In a word, via the ingeniously designed research platform, the efficiency enhancement of hydrogen spillover assisted by water molecules is achieved in MOF-801.

## Regional calculation of water-assisted hydrogen spillover in MOF-801
Measuring the efficiency of water-assisted hydrogen spillover is critical for understanding the spillover mechanism and exploring applications in catalysis. Herein, a spillover model was established for calculating the region of hydrogen spillover in Pt@MOF-801(wet). In this model, activated hydrogen atoms were generated from Pt NPs and transferred from the central to peripheral areas of Pt@MOF-801. As shown in Fig. 3a, Pt@MOF-801(wet) could be regarded as a core–shell structure with the spillover region (yellow part) as the core and the non-spillover

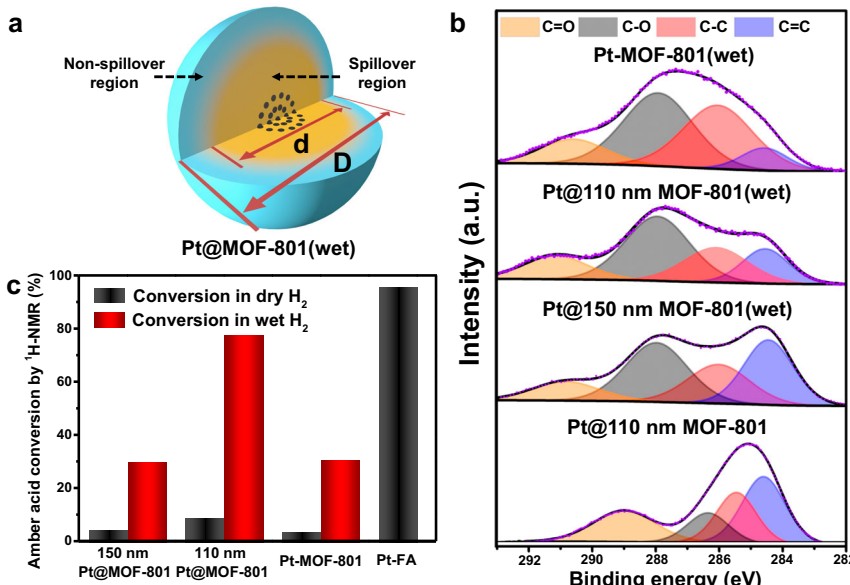

**Fig. 3 | Simulation and verification of the hydrogen spillover region in MOF-801 in wet H₂ thermal treatment. a** Core–shell spillover model of Pt@MOF-801(wet) including spillover region as core and non-spillover region as shell. **b** C 1*s* XPS spectra of Pt@110 nm MOF-801, Pt@150 nm MOF-801(wet), Pt@110 nm MOF-801(wet), and Pt-MOF-801(wet). **c** Amber acid conversion of Pt@150 nm MOF-801, Pt@110 nm MOF-801, and Pt-MOF-801 after dry or wet H₂ thermal treatment. Amber acid conversion of Pt-FA after dry H₂ thermal treatment.

region (blue part) as the shell. Due to the uniform spherical morphology of Pt@MOF-801 particles, the proportion of the spillover region to the whole region corresponded to the conversion rate of ligands in Pt@MOF-801(wet), which could be summarized as Eq. (1).

$$\alpha = A \div V = (d \div D)^3 \qquad (1)$$

Here, $\alpha$ is the conversion rate of the ligand, $A$ is the volume of the spillover region, $V$ is the total volume, $d$ is the diameter of the spillover region, and $D$ is the overall diameter of the MOF particle. The ligand conversions were measured by proton nuclear magnetic resonance ($^1$H-NMR) (Fig. 3c and Supplementary Fig. 21). The conversion of Pt-FA was 96%, which suggested fumaric acid could be hydrogenated efficiently under 150 °C in H₂. Furthermore, the conversions of Pt@150 nm MOF-801(dry), Pt@110 nm MOF-801(dry), and Pt-MOF-801(dry) were calculated to be 4.2%, 8.7%, and 3.5%, respectively, which indicated that only a small part of the ligands around Pt NPs were hydrogenated under dry H₂ treatment. The ligand conversions of Pt@150 nm MOF-801(wet), Pt@110 nm MOF-801(wet), and Pt-MOF-801(wet) were 30.0%, 77.8%, and 30.6% (Fig. 3c). Therefore, the spillover region diameters of Pt@150 nm MOF-801(wet) and Pt@110 nm MOF-801(wet) could be calculated by Eq. (1) as around 99 nm and 104 nm, respectively. To further support the results of the spillover region, X-ray photoelectron spectroscopy (XPS) was used for surface hydrogenation detection[54,55]. The area ratios of C–C to C=C in Pt@110 nm MOF-801, Pt@150 nm MOF-801(wet), Pt@110 nm MOF-801(wet), and Pt-MOF-801(wet) were 0.76, 0.76, 1.42, and 2.50 (Fig. 3b). The area ratio witnessed a boost in Pt-MOF-801(wet), suggesting a higher reduction degree of the ethylenic bond. At the same time, the area ratio of Pt@110 nm MOF-801(wet) was lower than that of Pt-MOF-801(wet) but higher than that before treatment, which implied that partial hydrogenation occurs on the surface of Pt@110 nm MOF-801(wet) and the activated hydrogen atoms diffuse exactly on the surface of Pt@110 nm MOF-801(wet). The ratio in Pt@150 nm MOF-801(wet) was close to the ratio before treatment, which suggested that the activated hydrogen atom hardly reaches the surface of Pt@150 nm MOF-801(wet) after wet H₂ thermal treatment. The influence of treatment time on the

hydrogen spillover was investigated by thermally treating Pt@150 nm MOF-801 in wet H₂ for 30 min, 2 h, and 4 h, which, respectively, exhibited 14.5%, 32.8%, and 34.3% for ligand conversion (Supplementary Fig. 23). These results indicated the maximum efficiency existence of hydrogen spillover for MOF-801 in the wet H₂ treatment, which hardly increases with treatment time extension. In addition, with a higher H₂ pressure and humidity in a closed high-pressure reactor with 60 mL in volume (Supplementary Fig. 22), the Pt@150 nm MOF-801 exhibited a larger spillover region diameter, which basically spanned the whole MOF particles (named as Pt@150 nm MOF-801 under high pressure, Supplementary Figs. 24–27 and Supplementary Discussion). These results evidenced that our strategy benefits the regulation of spillover efficiency and provides guidance for potential catalytic applications.

## Process analysis of water-assisted hydrogen spillover in MOF-801

In order to elucidate the process of hydrogen spillover, deuterium labeling experiments were adopted to track the trajectories of water and hydrogen molecules to ligands by mass spectrometry (MS) (Supplementary Fig. 28). After being treated in H₂O-containing D₂ or D₂O-containing H₂, fumaric acid in MOF-801 was obviously deuterated, which may be attributed to the hydrogen–deuterium exchange with the assistance of water[56,57] and μ₃-OH groups[58] assistance. At the same time, the hydrogenation of fumaric acid was hardly observed in MOF-801, probably due to the absence of Pt NPs for H₂ splitting (Fig. 4a, b). Interestingly, after being treated in H₂O-containing D₂, the ligand in Pt@110 nm MOF-801 would be hydrogenated to [D₂]-amber acid, [D₃]-amber acid or [D₄]-amber acid with no amber acid or [D]-amber acid (Fig. 4c). The above analysis demonstrated that H₂ splitting can occur on Pt NPs, and the activated hydrogen atoms can diffuse to the ligand of MOF-801. In contrast, after being treated in D₂O-containing H₂, the ligand in Pt@110 nm MOF-801 would be hydrogenated to amber acid or [D]-amber acid rather than more deuterated amber acid (Fig. 4d). The different deuterated results led to a finding that the activated hydrogen atoms, which was split from the H₂, could hydrogenate ligands and exchange with the D₂O. The above analysis indicates that the migration process of hydrogen spillover includes H₂ splitting on Pt

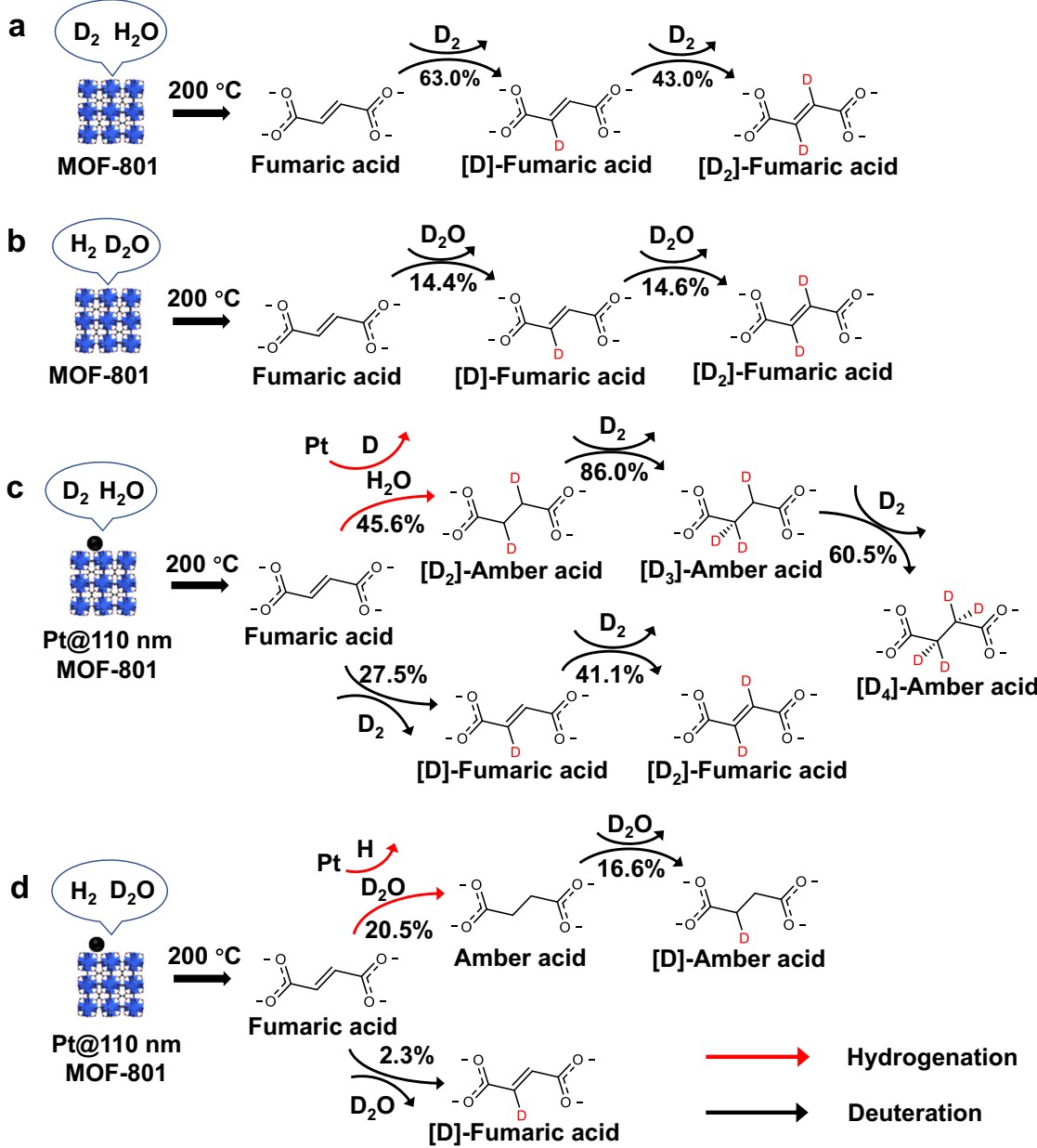

**Fig. 4 | Deuterium labeling experiments for characterizing the effect of water in water-assisted hydrogen spillover.** The ligand conversion of MOF-801 after thermal treatment in **a** $H_2O$-containing $D_2$ and **b** $D_2O$-containing $H_2$. The ligand conversion of Pt@110 nm MOF-801 after thermal treatment in **c** $H_2O$-containing $D_2$ and **d** $D_2O$-containing $H_2$.

NPs, and activated hydrogen atom diffusion to the ligand of MOF-801 over $H_2O$ molecules.

In the process of water-assisted hydrogen spillover in MOF-801, ligand hydrogenation and MOF structure collapse were observed, which raised questions about their influence on hydrogen spillover. During the hydrogen spillover process in MOF-801, hydrogen spillover is considered as the source of ligand hydrogenation and occurs much earlier than ligand hydrogenation and MOF structure collapse. Studies using scanning tunneling microscopy (STM) indicated that hydrogen spillover would reach equilibrium in a short period of time, typically completing the H diffusion process in a few seconds[39]. Meanwhile, the collision theory of chemical reactions indicated that during the H diffusion, the ligand would undergo multitudinous collisions of hydrogen atoms for hydrogenation, so that the hydrogenation in MOFs required a longer time, which was affected by energy barriers and probabilities. At the same time, the collapse of MOF structure was

caused by ligand hydrogenation and occurred later than hydrogen spillover. Therefore, at the initial stage of the reaction, the hydrogen spillover in MOF-801 would not be limited by MOF structure collapse due to the earlier occurrence of hydrogen spillover. In order to demonstrate the concept, we studied the sequence of hydrogen spillover in MOFs and MOF structure collapse. In the case of Pt@110 nm MOF-801 in wet $H_2$, we observed slight discoloration of tungsten oxide after 5 min of thermal treatment (Supplementary Fig. 29). This discoloration became significant after 20 min, despite only 7.6% ligand conversion (Supplementary Fig. 30). These results confirmed that water-assisted hydrogen spillover occurs earlier than the collapse of the MOF-801 structure. The PXRD pattern (Supplementary Fig. 31) and nitrogen adsorption–desorption isotherms (Supplementary Figs. 32, 33) also supported these findings. By contrast, when dry $H_2$ was used in the thermal treatment of Pt@110 nm MOF-801, there was hardly any discoloration of tungsten oxide even after 2 h. This

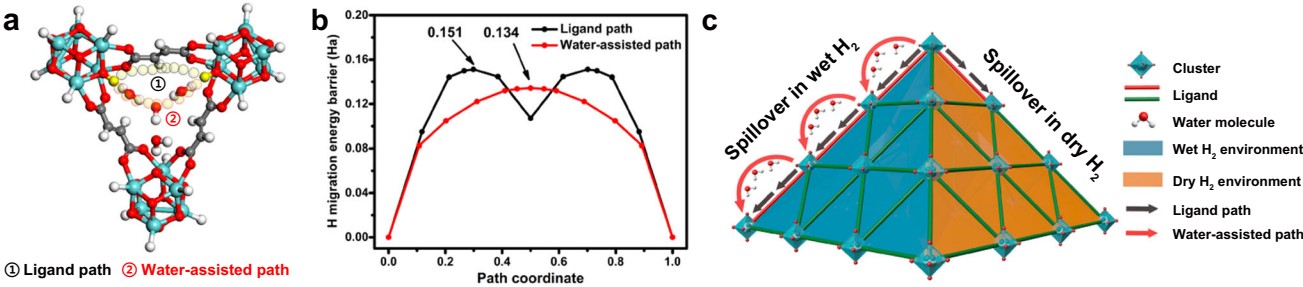

**Fig. 5 | Proposed mechanism of H migration with energy barriers calculation in MOF-801. a** Schematic representation of H migration path in MOF-801. **b** The calculated H migration energy barriers of two paths in MOF-801. **c** Scheme of possible modes of hydrogen spillover in dry or wet $H_2$ in MOF-801.

suggested that with the assistance of water molecules, hydrogen spillover seems to cover the region completely before structural damage takes place. Therefore, the hydrogen spillover region can be estimated by analyzing the ligand hydrogenation.

### Mechanism of water-assisted hydrogen spillover in MOF-801

Understanding the mechanism of water-assisted hydrogen spillover in MOF-801 becomes our top concern. To further understand the mechanism, the hydrogen spillover model was established according to the structure of MOF-801 and the water location in the channel. Generally, the water absorption in MOF-801 belongs to reversible cluster adsorption[53]. In our experiments, when the input wet $H_2$ (28 °C and 80% RH) was heated to 200 °C, the relative humidity of the input wet $H_2$ would be reduced to around 0.19% RH according to the Antoine formula for water as Eq. (2). The 0.19% RH is a quite low humidity condition, which corresponds to the water location in the MOF-801 channel under low uptake[53]. According to the reported literature[58], the water molecules tend to adsorb within the two tetrahedral pores close to the $\mu_3$-OH groups (Supplementary Fig. 34) of the zirconium-oxide clusters as the primary adsorption sites through hydrogen bonds. Such adsorbed water molecules would be further connected to form water molecular clusters, which is consistent with the water molecule distribution in the tetrahedral cavity of powder MOF-801 as reported[53]. After confirming the model structure, another primary concern was the diffusion of activated hydrogen atoms in MOFs. As a crucial part of MOF-801, zirconium-oxide clusters are an indispensable part of the traditional hydrogen spillover path in MOFs, the diffusion of which is not the kinetical rate-limiting step with a lower spillover barrier than that of ligands in the spillover process[59]. Hence, based on the analysis of the previous experimental results and the model structure, two possible spillover paths were proposed between two clusters including the first path via the ligand and the second path via the water chain, and calculated by DFT method with DMol3 program (Fig. 5a). Specifically, three clusters were cut from the optimized periodic structure and terminated with hydrogen atoms, which were utilized for hydrogen spillover calculation. In the first ligand path (Supplementary Fig. 35a), the activated hydrogen atoms could overcome the high migration energy barrier of 0.151 Ha for migration to $sp^1$ C−C single bond first, then migrate to $sp^2$ C=C bond for energy release of 0.044 Ha, and finally overcome the migration energy barrier for further migration to $sp^1$ C−C single bond (Fig. 5b). The two migrations with high energy barriers limited the hydrogen spillover for the first path, which was consistent with theoretical studies of MOF hydrogen spillover mechanisms[60]. For the second path (Supplementary Fig. 35b), spillover H atoms from the MOF clusters may be protonated and transferred to the water clusters. And the protonated water clusters ($[H_{2n+1}O_n]^{\delta+}$) were formed and regarded as spillover H receptors to the MOF[42]. Due to the transient formation of diverse complex transition states of water clusters, the hydrogen atoms in protonated water clusters were rearranged in bonding topologies in confined nanochannels, which may lead to a water-assisted spillover path (Supplementary Fig. 36)[61,62]. Therefore,

the second water-assisted path had a maximum migration energy barrier of 0.134 Ha and possessed a kinetic advance compared with the first path. Nevertheless, according to the calculation of the Gibbs free energy change as Eq. (3), $\Delta E$ of hydrogenation energy in MOF-801 was −0.092 Ha. This negative value in energy meant that the ligand could be hydrogenated by the activated hydrogen atoms (Supplementary Fig. 37). Therefore, under dry $H_2$ treatment, diffusion only relies on the first path (cluster–ligand–cluster), and the high diffusion energy barrier of this path results in obstructive hydrogen spillover. Under wet $H_2$ treatment, not only can the first path work, but the second path also allow superior diffusion of activated hydrogen atoms between clusters with a lower energy barrier. The synergistic effect of the two paths may be the key reason behind the efficiency enhancement of water-assisted hydrogen spillover in MOF-801 (Fig. 5c).

### Extensibility of water-assisted hydrogen spillover in more MOFs and COFs

Since water absorption widely exists in porous crystalline materials[63], these interesting findings of water-assisted hydrogen spillover in MOF-801 inspire our imagination to extend this method to more MOFs or even COFs. Here, UiO-66 and UiO-67, which had the same *fcu* topology but different ligands as MOF-801, and ZIF-8, ZIF-67, HKUST-1, and Fe-MIL-53 with diverse metal clusters, were taken into consideration. Accordingly, Pt@UiO-66 (Supplementary Fig. 38a), Pt@UiO-67 (Supplementary Fig. 38b), Pt@ZIF-8 (Supplementary Fig. 38c), Pt@ZIF-67 (Supplementary Fig. 38d), Pt@HKUST-1 (Supplementary Fig. 38e), and Pt@Fe-MIL-53 (Supplementary Fig. 38f) were synthesized by controlled nanoparticle encapsulation method[64]. Similarly, Pt/TAPT-DHTA (Supplementary Fig. 38g) and Pt/TpPa-1 (Supplementary Fig. 38h) were synthesized by reduction of Pt nanoparticles in presynthesized COFs. These Pt@MOFs or Pt/COFs were, respectively, mixed with tungsten oxide and processed in dry or wet $H_2$ at 200 °C (Fig. 6). These mixtures relatively showed more obvious discoloration under wet $H_2$ thermal treatment than those under dry $H_2$ thermal treatment. The activated hydrogen atoms on the surface of these supports, resulting in tungsten oxide discoloration, were generated through hydrogen spillover, which can be enhanced with the assistance of absorbed water molecules. After dry or wet hydrogen thermal treatment, Pt@UiO-66, Pt@UiO-67, Pt@ZIF-8, and Pt@ ZIF-67 maintained similar PXRD patterns (Supplementary Fig. 39), nitrogen adsorption–desorption curves, and pore size distributions (Supplementary Fig. 40). These data indicated the negligible influence of hydrogen spillover on their structure, which reflected that UiO-66, UiO-67, ZIF-8, and ZIF-67 are relatively stable for hydrogen spillover. Furthermore, Pt@HKUST-1, Pt@Fe-MIL-53, Pt/TAPT-DHTA, and Pt/TpPa-1 exhibited crystal structure damage (Supplementary Fig. 39) and declined BET surface area (Supplementary Fig. 41) after dry or wet hydrogen thermal treatment. On the one hand, according to the obvious influence of hydrogen spillover on their structure,

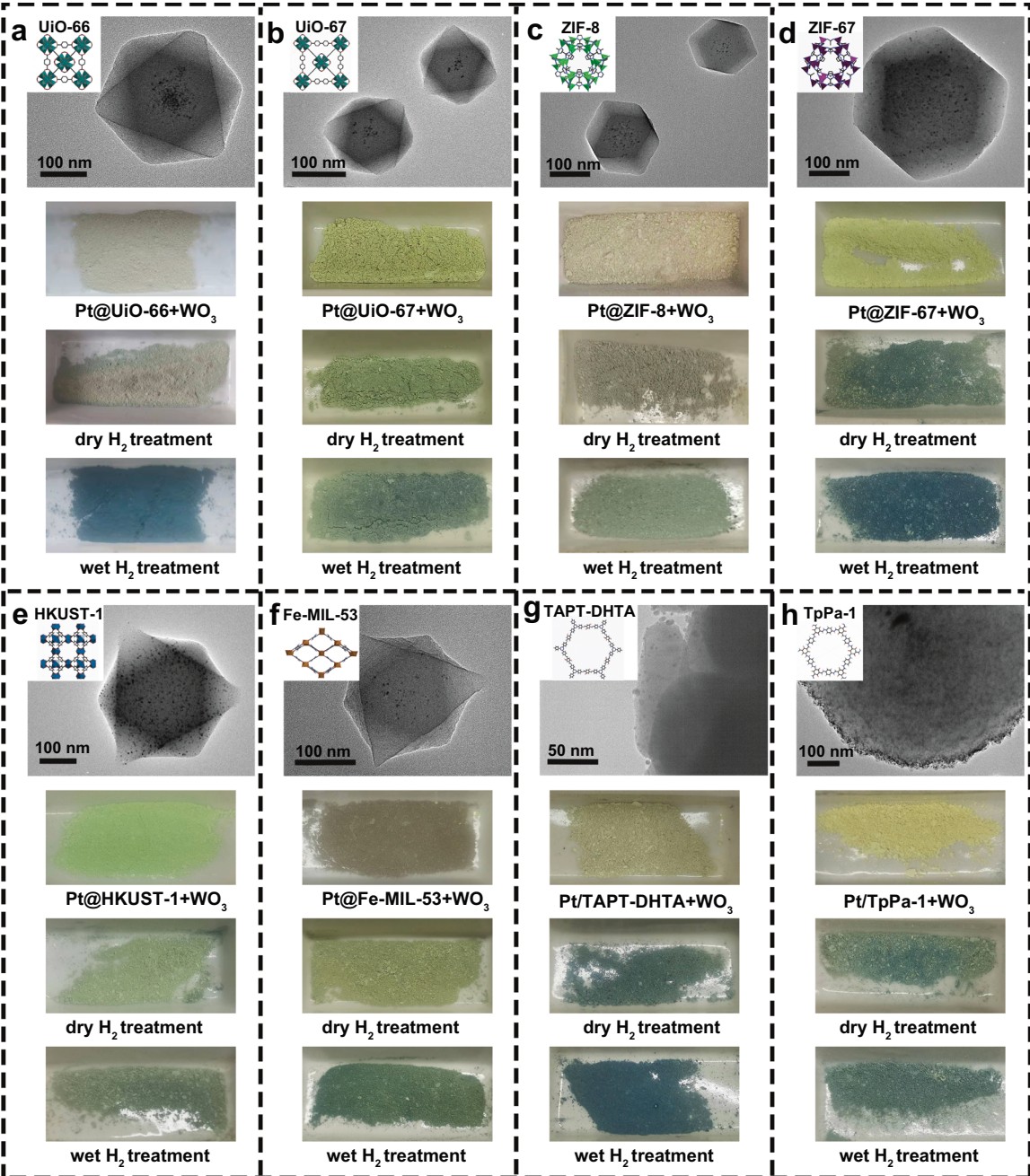

**Fig. 6 | Water-assisted hydrogen spillover in more MOFs and COFs.** The structure information, TEM images and photographs of WO$_3$ and samples including **a** UiO-66, **b** UiO-67, **c** ZIF-8, **d** ZIF-67, **e** HKUST-1, **f** Fe-MIL-53, **g** TAPT-DHTA, and **h** TaPa-1 after H$_2$ treatment in different humidities.

Pt@HKUST-1, Pt@Fe-MIL-53, Pt/TAPT-DHTA, and Pt/TpPa-1 were relatively unstable for hydrogen spillover, which may be caused by the reducible metal clusters or hydrogenable ligands. Interestingly, the damage to Pt@HKUST-1, Pt@Fe-MIL-53, and Pt/TAPT-DHTA was more pronounced after wet hydrogen thermal treatment compared to dry hydrogen thermal treatment. Such a universal strategy of water-assisted hydrogen spillover can be extended not only to stable MOFs for the design of porous crystalline catalysts, but also to unstable porous materials for the preparation of derivatives.

## Antitoxic nitro hydrogenation by water-assisted hydrogen spillover

In some catalytic reaction systems, it is a common phenomenon that trace toxic impurities in reactants may significantly damage the

activity and selectivity of the catalyst. In general, pore confinement or encapsulation of active centers by porous material is an effective method to improve catalyst stability and prevent toxicant by unique channels[65]. However, in many cases, the sizes of reactants and impurities are too close to be screened by channels, resulting in catalyst poisoning or preventing the reactants from touching the active centers. Fortunately, hydrogen spillover may be a potential way to achieve reactant hydrogenation by activated hydrogen atoms while avoiding the poison of active centers, in particular in MOF-based catalysts by our hydrogen spillover enhancement strategy. Herein, the 2,6-dimethylnitrobenzene (4.02 × 8.32 × 8.9 Å, Supplementary Fig. 42) and 4,4′-dithiodipyridine (DTDP) (5.30 × 6.46 × 12.88 Å, Supplementary Fig. 43) with both larger molecular sizes than the window of ZIF-8 (3.4 Å) were selected as model reactant for nitro hydrogenation, and model toxicant due to the high toxicity of sulfur and pyridine compounds to

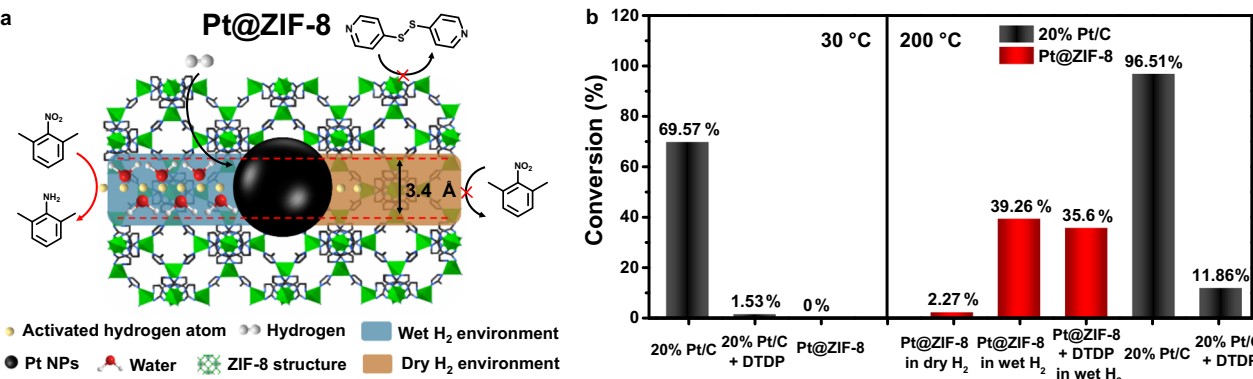

**Fig. 7 | Antitoxic nitro hydrogenation at the outside of Pt@ZIF-8 by water-assisted hydrogen spillover. a** Scheme of nitro hydrogenation by Pt@ZIF-8 under dry or wet $H_2$ environments. **b** Nitro hydrogenation conversion at 30 °C including commercial 20% Pt/C, commercial 20% Pt/C with DTDP and Pt@ZIF-8, respectively. And nitro hydrogenation conversion at 200 °C including Pt@ZIF-8 in dry $H_2$, Pt@ZIF-8 in wet $H_2$, Pt@ZIF-8 with DTDP in wet $H_2$, commercial 20% Pt/C, and commercial 20% Pt/C with DTDP, respectively.

noble metal centers, respectively (Fig. 7a). Owing to the steric effect, Pt@ZIF-8 exhibited no activity on 2,6-dimethylnitrobenzene hydrogenation in ethyl acetate at 30 °C for 12 h, which proved that the macromolecular reactants are inaccessible to Pt NPs in MOFs. Meanwhile, the commercial 20% Pt/C catalyst had an appreciable conversion (69.57%) for 2,6-dimethylaniline under similar conditions; however, it decreased to 1.53% after being poisoned by the introduction of DTDP. Furthermore, in order to hydrogenate macromolecular reactant at the outside of MOFs by hydrogen spillover, hydrogenation was investigated in dry or wet $H_2$ conditions at 200 °C without additional solvent for 8 h. The Pt@ZIF-8 had a low conversion (2.27%) in dry $H_2$, which should be attributed to the low efficiency of hydrogen spillover in ZIF-8. On the contrary, the conversion in wet $H_2$ had an obvious improvement (39.26%) due to the efficiency enhancement of hydrogen spillover by our strategy (Supplementary Fig. 44). More importantly, Pt@ZIF-8 showed similar conversion (35.60%) with the existence of toxicant, mainly due to the protection of MOF and the improved hydrogen spillover. Meanwhile, the commercial 20% Pt/C catalyst displayed a significant drop in conversion from 96.51 to 11.86% at 200 °C. We believed that the conversion improvement and antitoxicity based on the design of hydrogen spillover in MOFs represent a promising beginning and highlight the potential of this approach in catalytic research. The active site transferring from Pt NPs to MOFs also has great potential for extending the application of MOF-based catalysts due to their adjustable channel environment, designable MOF electronic state, and distinctive metal clusters and ligands as active centers.

## Discussion

In summary, we develop a water-assisted hydrogen spillover strategy for the efficiency enhancement of hydrogen spillover in MOFs. To explore this strategy, a research platform for evaluating hydrogen spillover is designed via characterization of ligand conversion and structural changes as detectors in MOF-801. The hydrogen spillover is proven to be undetectable under dry $H_2$ treatment due to the high energy barrier in MOFs. Meanwhile, water molecules provide a spillover path to enhance hydrogen spillover in Pt@MOF-801 at 200 °C, spanning a region of around 100 nm in diameter at atmospheric pressure and even a larger region under higher humidity and pressure conditions. Furthermore, the deuterium-labeled experiments demonstrate the process including activated hydrogen atom generation on Pt NPs from $H_2$ splitting and diffusion on MOF-801 under water assistance. DFT calculations simulate the migration process of activated hydrogen atoms under water assistance and a spillover path with a lower energy barrier than the traditional ligand path. The synergy of these two paths promotes hydrogen diffusion between zirconium-oxide clusters, largely accounting for the efficiency enhancement of hydrogen spillover under wet $H_2$ treatment. More importantly, such a strategy can be applied to more MOF and COF materials, which suggests broad application prospects. The establishment of the research platform, the revelation of spillover mechanism and the strategy of capacity improvement for hydrogen spillover in MOFs are expected to open up new horizons for a deeper understanding of hydrogen spillover in porous crystalline materials and designing more efficient hydrogenation or dehydrogenation composite catalysts.

## Methods
### Materials

All commercially available reagents and solvents were used as received without further purification. $ZrOCl_2$ and $D_2O$ were bought from Energy Chemical. Fumaric acid and $FeCl_3·6H_2O$ was purchased from Alfa Aesar. $Zn(NO_3)_2·6H_2O$, $Co(NO_3)_2·6H_2O$, $Cu(NO_3)_2·3H_2O$, 2-methylimidazole, $K_2PtCl_4$, polyvinyl pyrrolidone, terephthalic acid, dimethyl sulfoxide-d6 (DMSO-d6), 2,6-dimethylnitrobenzene, 4,4′-dithiodipyridine and biphenyl-4,4′-dicarboxylate were bought from Sigma Aldrich. $ZrCl_4$ was purchased from Acros Organics.

### Synthesis of Pt nanoparticles

Pt nanoparticles were synthesized according to a reported method[26]. 50 mg $K_2PtCl_4$ was dissolved in 20 mL deionized water and 133 mg polyvinyl pyrrolidone (PVP, MW = 29,000) was dissolved in 180 mL of methanol. The two solutions were mixed under stirring at room temperature for 10 min. The Pt nanoparticles were formed after refluxing for 3 h and being evaporated by rotary evaporation. After evaporation in a vacuum drying oven for 3 days at 60 °C, the Pt nanoparticles were washed over 10 times with chloroform and n-hexane to remove excess free PVP and dispersed in 60 mL N,N-dimethylformamide (DMF), ethanol or methanol as a stock solution.

### Synthesis of MOF-801

According to a reported method[49], 322 mg $ZrOCl_2·8H_2O$, 580 mg fumaric acid, and 15 mL acetic acid (HAc) were dissolved in 70 mL DMF in a glass bottle. After 15 min ultrasonic treatment, the vessel was placed in an oven at 90 °C for 18 h. The MOF-801 was centrifuged at 11,627 x*g* for 5 min, washed three times with ethanol and dried at 80 °C.

### Synthesis of Pt@110 nm MOF-801 and Pt@150 nm MOF-801

According to the controlled nanoparticle encapsulation method[66], 322 mg $ZrOCl_2·8H_2O$, 580 mg fumaric acid, 25 mL HAc and 10 mL as-synthesized Pt nanoparticles in DMF were dissolved in 95 mL DMF in a

glass bottle. After ultrasonic treatment for 15 min, the vessel was placed in an oven at 90 °C for 9 h to obtain Pt@110 nm MOF-801 and 18 h to obtain Pt@150 nm MOF-801. The products were centrifuged at 11,627 x*g* for 5 min, washed three times with ethanol, and dried at 80 °C.

## Synthesis of Pt-MOF-801 and Pt-FA
240 mg as-synthesized MOF-801 was dispersed in 10 mL as-synthesized Pt nanoparticles of DMF solution. After stirring at room temperature for 3 h, Pt-MOF-801 was rotarily evaporated and dried at 80 °C. 300 mg fumaric acid was dispersed in 10 mL as-synthesized Pt nanoparticles of ethanol. After stirring at room temperature for 3 h, the Pt-FA was rotarily evaporated and dried at 80 °C.

## Synthesis of Pt@UiO-66
According to the reported method[67], 19 mg $ZrCl_4$, 13.5 mg terephthalic acid, 1.35 mL HAc, and 0.8 mL as-synthesized Pt nanoparticles in DMF were dissolved in 10 mL DMF in a glass bottle. After 15 min ultrasonic treatment, the vessel was placed in an oven at 120 °C for 18 h. The Pt@UiO-66 was centrifuged at 11,627 x*g* for 5 min, washed three times with ethanol, and dried at 80 °C.

## Synthesis of Pt@UiO-67
According to the reported method[49], 363.2 mg biphenyl-4,4′-dicarboxylate (BPDC), 16 mL HAc, and 8 mL as-synthesized Pt nanoparticles in DMF were dissolved in 200 mL DMF at 90 °C in a glass bottle. After cooling to room temperature, 193.4 mg $ZrOCl_2 \cdot 8H_2O$ was dissolved in the solution. After 15 min ultrasonic treatment, the vessel was placed in an oven at 90 °C for 9 h. The Pt@UiO-67 was centrifuged at 11,627 x*g* for 5 min, washed three times with ethanol, and dried at 80 °C.

## Synthesis of Pt@ZIF-8
According to the reported method[64], 30.8 mg 2-methylimidazole and 56 mg $Zn(NO_3)_2 \cdot 6H_2O$ were dissolved in 15 mL and 8 mL methanol, respectively. 0.33 mL as-synthesized Pt nanoparticles in methanol was added to the mixture of the two previous solutions under stirring for 10 min. After standing for 12 h at room temperature, the Pt@ZIF-8 was centrifuged at 11,627 x*g* for 5 min, washed three times with methanol, and dried at 80 °C.

## Synthesis of Pt@ZIF-67
260 mg 2-methylimidazole and 115 mg $Co(NO_3)_2 \cdot 6H_2O$ were dissolved in 5 mL methanol, respectively. 1 mL as-synthesized Pt nanoparticles in methanol was added to the mixture of these solutions under stirring for 10 min. After standing for 2 h at room temperature, the Pt@ZIF-67 was centrifuged at 11,627 x*g* for 5 min, washed three times with methanol, and dried at 80 °C.

## Synthesis of Pt@HKUST-1
According to the reported method[68], 460 mg $Cu(NO_3)_2 \cdot 3H_2O$ and 222 mg benzene-1,3,5-tricarboxylic acid were dissolved in a 10 mL glass bottle with 2 mL ethanol, 2 mL deionized water, and 2 mL as-synthesized Pt nanoparticles in DMF. After 15 min ultrasonic treatment, the vessel was placed in an oven at 85 °C for 1 h. The Pt@HKUST-1 was centrifuged at 11,627 x*g* for 5 min, washed three times with ethanol, and dried at 80 °C.

## Synthesis of Pt@Fe-MIL-53
According to the reported method[50], 90 mg $FeCl_3 \cdot 6H_2O$ and 135 mg 2-aminoterephthalic acid were dissolved in a 40 mL glass bottle with 30 mL DMF and 3 mL as-synthesized Pt nanoparticles in DMF. After 15 min ultrasonic treatment, the vessel was placed in an oven at 120 °C for 6 h. The Pt@Fe-MIL-53 was centrifuged at 11,627 x*g* for 5 min, washed three times with ethanol, and dried at 80 °C.

## Synthesis of Pt/TAPT-DHTA
28 mg 1,3,5-tris(4-aminophenyl)triazine (TAPT) and 19 mg 2,5-dihydroxyl-terephthalaldehyde (DHTA) were suspended in a 10 mL glass bottle with 1.6 mL 1,3,5-trimethylbenzene, 0.4 mL 1,4-dioxane, and 0.05 mL HAC. After 30 min ultrasonic treatment, the vessel was placed in an oven at 120 °C for 3 days. The TAPT-DHTA was centrifuged at 11,627 x*g* for 5 min, washed three times with ethanol, and dried at 80 °C. 50 mg TAPT-DHTA and 25 mg platinum(II) acetylacetonate were suspended in 2 mL toluene. After ultrasonic treatment for 15 min and standing for 24 h, the mixture was centrifuged at 11,627 x*g* for 5 min, washed with ethanol for one time, and calcined in $H_2$ at 200 °C for 2 h.

## Synthesis of Pt/TpPa-1
63 mg triformylphloroglucinol (Tp) and 48 mg paraphenylenediamine (Pa-1) were suspended in 10 mL glass bottle with a mixture of 3 mL dimethylacetamide and 3 mL o-dichlorobenzene. After 30 min ultrasonic treatment, the vessel was placed in an oven at 120 °C for 3 days. The TpPa-1 was centrifuged at 11,627 x*g* for 5 min, washed three times with ethanol, and dried at 80 °C. 50 mg TpPa-1 and 25 mg platinum(II) acetylacetonate were suspended in 2 mL toluene. After ultrasonic treatment for 15 min and standing for 24 h, the mixture was centrifuged at 11,627 x*g* for 5 min, washed once with ethanol, and calcined in $H_2$ at 200 °C for 2 h.

## Materials characterizations
PXRD patterns of samples were recorded with a Bruker AXS D8 Advance diffractometer using nickel-filtered Cu Kα radiation ($\lambda = 1.5406$ Å). MS was performed with a Thermo Scientific Q Exactive in negative ion mode using an electron spray ionization (ESI) ion source from 50 to 200. To prepare the samples, 20 mg of dried MOF powders were digested and dissolved in a mixture of methanol (2 mL) and hydrofluoric acid (100 µL) with sonication. After adding 1 mL acetone and centrifuging at 465 x*g*, the samples were collected after drying at 100 °C. Scanning electron microscopy (SEM) images were taken by a JEOL JSM-7600 with an accelerating voltage of 10 kV. TEM images were taken by JEOL JEM-2100 Plus at an accelerating voltage of 200 kV. The temperature and humidity were detected with a YOWEXA humidity recorder SSN-22 in gas inlet of tube furnace. Nitrogen adsorption–desorption isotherms of powder samples were measured with a Micromeritics ASAP 2460 adsorption apparatus at 77 K up to 1 bar. Before starting the adsorption measurements, each sample was activated by heating under vacuum at 150 °C for 12 h. The pore textural properties including BET surface area, pore volume, and pore size were obtained by analyzing nitrogen adsorption–desorption isotherms with the DFT method. $^1$H NMR analysis was detected with a Bruker spectrometer (400 MHz). 30 mg of dried MOF powders were digested and dissolved in a mixture of DMSO-d6 (570 µL) and hydrofluoric acid (80 µL) with sonication[69]. TGA was performed on a Q500 TGA (TA Instruments) under nitrogen gas flow at 5 °C·min$^{-1}$ from 30 °C to 600 °C. TPR was performed on a Quantachrome ChemBET TPR/TPD chemisorption analyzer under hydrogen gas flow at 10 °C·min$^{-1}$ from 30 °C to 500 °C. Gas chromatography (GC) spectra were recorded on an Agilent Technologies 7890B.

## Thermal treatment of samples
Samples including Pt@150 nm MOF-801, Pt@110 nm MOF-801, Pt-MOF-801, other Pt@MOFs and Pt/COFs were calcined in a tube furnace via a gas-washing bottle, which was loaded with CaO powder desiccant for dry gas or water for wet gas. In a typical hydrogen spillover process, the temperature increased from 30 °C to 200 °C for 20 min and was held at 200 °C for 120 min in $N_2$ and $H_2$ successively.

 

## Relative humidity estimation

Relative humidity conversion of 80% RH from 28 °C to 200 °C by the Antoine formula for water.

$$\ln(P) = 9.3876 - 3{,}826.36 \times (T - 45.47) \tag{2}$$

$P$---saturated vapor pressure (MPa)

$T$---temperature (K, 290 < K < 500)

Thus, the saturated vapor pressures of water at 28 °C and 200 °C are 3.78 KPa and 1553.98 KPa, respectively. The absolute vapor pressure is 3.024 KPa (3.78 KPa × 80%). The relative humidity at 200 °C is 0.19% RH (3.024 KPa÷1553.98 KPa).

## Hydrogen spillover under high $H_2$ pressure and humidity

Hydrogen spillover under higher $H_2$ pressure and humidity was investigated in a closed high-pressure reactor with a 60-mL volume (Supplementary Fig. 22). Pt@110 nm MOF-801 and Pt@150 nm MOF-801 were processed in the reactor with 5 mL water and 0.4 MPa $H_2$ pre-filling at 200 °C for 2 h.

## DFT calculations

All calculations have been carried out using the DMol3 program[70,71] based on DFT with GGA-PBE[72] for exchange and correlation potential. We used DFT semicore pseudopotential with a double numerical basis set plus polarization functions (DNP). Convergence in energy, force, and displacement were set as $10^{-5}$ Ha, 0.001 Ha/Å, and 0.005 Å, respectively. The core treatment was set to effective core potentials (ECP). The space cut off radius maintained at 4.4 Å. All transition states were located via the complete LST method[73] as implemented in DMol3 package.

The hydrogenation energy on MOF-801 is calculated by the following equation:

$$\Delta E = E_{(MOF+H)} - E_{(MOF)} - 0.5 \times E(H_2) \tag{3}$$

Where $E_{(MOF+H)}$ represents the energy of MOF-801 with the adsorbed H, $E_{(MOF)}$ is the energy of MOF-801, and $E(H_2)$ is the energy of hydrogen molecule. The corresponding adsorption structure is shown in Supplementary Fig. 37.

## Catalysis of 2,6-dimethylnitrobenzene hydrogenation

For 2,6-dimethylnitrobenzene hydrogenation at 30 °C, 20 mg Pt@ZIF-8, or 10 mg commercial 20% Pt/C catalyst was dispersed in 0.5 mL 2,6-dimethylnitrobenzene and 3 mL ethyl acetate. And then 5 mg DTDP was provided additionally to investigate catalyst antitoxicity. The hydrogenation took place in a closed high-pressure reactor with a 60-mL volume and initial 0.4 MPa hydrogen at 30 °C for 12 h. The reaction products were directly monitored by GC.

For 2,6-dimethylnitrobenzene hydrogenation at 200 °C, 20 mg Pt@ZIF-8 was degassed at 180 °C for 12 h under vacuum and dispersed in 0.5 mL 2,6-dimethylnitrobenzene. In addition, 0.2 mL water was provided for humidity increase, and 5 mg DTDP was provided for investigating catalyst antitoxicity. Through inductively coupled plasma emission testing, the Pt ratio in Pt@ZIF-8 was calculated to be around 1.45%. And the Pt content in 20 mg Pt@ZIF-8 was equivalent to that in -1.5 mg commercial 20% Pt/C catalyst. To compare catalyst antitoxicity, 1.5 mg commercial 20% Pt/C catalyst was dispersed in 0.5 mL 2,6-dimethylnitrobenzene with and without additional 5 mg DTDP at 200 °C, respectively. The hydrogenation was performed in a closed high-pressure reactor with a 60-mL volume and initial 0.4 MPa hydrogen at 200 °C for 8 h. The reaction products were directly monitored by GC.

## Data availability

The authors declare that all the data supporting the findings of this study are available within the paper and its Supplementary Information or from the corresponding author on request.

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

## Acknowledgements

This work was supported by the National Natural Science Foundation of China (22175030, 21727808, 21971114, 62288102, and 21908105), the Open Project of State Key Laboratory of Supramolecular Structure and Materials (sklssm 202205), the Jiangsu Provincial Founds for Natural Science Foundation (BK20200090), the National Science Foundation for Distinguished Young Scholars (21625401), and the China Post-doctoral Science Foundation (2021M691552). Thanks to eceshi (www.eceshi.com) for MS analysis and Shiyanjia Lab (www.Shiyanjia.com) for XPS analysis. Special thanks to the instrumental from Analytical and Testing Center, Northeastern University.

## Author contributions

Z.G., Y.F., W.Z., and F.H. conceived and designed the project. Y.F., W.Z., and F.H. supervised the project and led the collaboration efforts. Z.G. synthesized and characterized the samples, conducted the spillover mechanism analysis and drafted the manuscript. M.L. performed the PXRD measurement. C.C. performed the $^1$H-NMR measurement. X.Z. and C.L. performed the TPR and TGA measurement. Y.Y. provided assistance on COF synthesis. M.L., R.S., and Y.S. helped the manuscript revision. Y.S. and S.Z. provided assistance on spillover mechanism analysis. All authors participated in the discussion and analysis of the manuscript.

## Competing interests

The authors declare no competing interests.
