## [Peer Review File · Nature Communications]

REVIEWER COMMENTS

Reviewer #1 (Remarks to the Author):

In the present study, the authors investigated hydrogen spillover in Pt@MOF-801 under both dry and wet H₂ conditions. The authors claimed that they discovered a new water-assisted hydrogen spillover pathway with a lower migration energy barrier. However, this statement is not true because the promotion effects of H₂O and other protic solvents on hydrogen spillover have been known for more than 50 years in the field of heterogeneous catalysis. In early 1970s, M. Boudart first reported that protic solvents can help the migration of surface H in a form of H⁺/e⁻ pairs via solvation (hydrogen bonding). Therefore, the authors' conclusions are not new at all. I agree that hydrogen spillover on MOF-based materials has not been systematically investigated thus far. The present study is a nice contribution, but has a significant logical flaw. The MOF-801 structure collapsed during the H spillover experiments. It is not clear whether the kinetics of ligand hydrogenation is limited by hydrogen spillover or the rates of MOF's structural collapse/ligand diffusion. The co-flow of water can not only facilitate hydrogen spillover but also the structural collapse of MOF and the diffusion of ligands onto the metal catalyst surface. Therefore, the MOF-801 system is not an ideal model system for rigorously investigating the kinetics of hydrogen spillover. The catalytic reaction studies were also carried out with little scientific rigor. Overall, I believe that this study cannot meet the high standard of Nature Communications, and should be submitted to other journals specialized in material chemistry.

Reviewer #2 (Remarks to the Author):

In this manuscript, Zhida Gu et al report on water-assisted hydrogen spillover in metal-organic frameworks. Hydrogen spillover phenomenon was detected by ligand conversion and structural changes as detectors in MOF-801. The mechanism was also discussed by deuterium-labeled experiments or DFT calculation. The enhanced hydrogen spillover by water can be applied to more MOFs or COFs materials. The results in this manuscript are surely interesting. However, there are several issues that the authors need to correspond. Once addressed the issues below, I like to reconsider the manuscript to be suitable for publication in Nature Communications.

1. In Fig. 2g, the structure of Pt@MOF-801 dramatically changed by wet H₂ thermal treatment at 30 °C. However, as shown in Fig. 2h, hydrogen consumption is negligible up to 250 °C. The authors are required to explain the reason.
2. In addition, in Fig. 2g, hydrogen consumption of Pt@MOF-801 is larger than that of Pt@MOF-801(wet). Why?
3. In Fig. 10S, the hydrogenation reaction of Pt@MOF-801(dry) seems to be progressing more than Pt@MOF-801(wet).
4. In Fig.7, the catalytic tests of 20% Pt/C should be performed under the same condition of Pt@ZIF-8 for appropriate comparison.
5. The authors only showed the color change before/after H₂ treatment to indicate efficiency promotion of hydrogen spillover of other MOFs or covalent organic framework materials (Fig. 6), but it is insufficient. The characterization of their materials after H₂ treatment is required e.g. XRD, N₂ sorption

Reviewer #3 (Remarks to the Author):

Hydrogen spillover is a very important interfacial phenomenon that has been well-documented in hydrogenation reactions. And, there have been many studies to report the existence of hydrogen spillover effect in noble metal-encapsulated MOFs or COFs. However, this research field is still in

its infancy and more efforts are needed to explore more effective detection methods of this phenomenon and to make clear its reaction mechanism. This study prepared a Pt nanoparticles@MOF-801 material to try to address some of the existing problems. In view of the literature, this work is not much new in terms of material preparation and subsequent mechanism verification. It is important that the authors chose a good MOF host where its ligands with the C=C double bond could serve a built-in detector for those activated hydrogen atoms by Pt, and that they reported a possibly new hydrogen spillover path assisted by water molecules adsorbing within this MOF host. However, I find that there are many important issues that needed to be addressed carefully, as follows:

1. The title is a little misleading and exaggerated, which ignored the key function of Pt.
2. The hydrogen migration distance should be expressed in terms of radius (namely shell thickness) rather than diameter centered on the location of these Pt atoms. It is misleading here. In this regard, Ref. 23 obviously reported a much larger region than this study, rather than "from imperceptible existence" the authors said here.
3. The samples' names in many Supplementary Figures are very chaotic. It is also difficult to distinguish "D" and "O" in Figure 4. Some of the key data occluded each other. In addition, the authors are suggested to name (and rename) the samples clearly at the very beginning. MOF-81 should be clearly introduced, including its compositions, structure, the position and amount of its adsorbed water, etc.. All in all, the present manuscript is hard to read and the authors should carefully refine and rearrange some of the contents. By the way, there are also some errors, like at Page 6 "Whist under wet H₂ treatment...", Figure S4 (the temperature should be 300 oC rather than 200 oC they discussed in the test "MOF-801 were tested respectively in wet H₂ and wet N₂ conditions at 200 °C"), and Figure 1 (which can't be found in the test).
4. Page 8: "The active site transfer from NPs to MOFs also has great potential on extending the application of MOF-based catalysts due to the adjustable channel environment, designable MOF electronic state and distinctive metal clusters and ligands as active centers." The active sites should still be Pt nanoparticles?
5. More clear TEM characterization results about Pt nanoparticles should be provided. And what are their sizes and weight ratios in each sample? Would these influence the performance analysis? What is the action of Zr cluster? Where are the μ 3-OH groups?
6. Are the TEM images in Figure 2d-f representative? More statistical images should be provided. And, to prove the reason for the adhesion phenomenon, the authors should prepare samples at a temperature below 188 oC.
7. The structure images for various MOFs are too large in Figure 6. The authors should clearly present their results rather than such information.
8. What is the evidence for the peak splitting of the XPS spectra? Reference should be added.

RESPONSE TO REVIEWERS' COMMENTS

Dear reviewers,

Thank you for reviewing our manuscript for publication on *Nature Communications*. We have gained a lot of improvement from the reviewers' comments, responded to the substantial concerns posed with more convincing experiments, revised the manuscript carefully based on the valuable suggestions and made relevant changes. All the changes of the manuscript are marked in highlight. The point-by-point responses to comments are listed as follows.

Reviewer 1: In this study, the author studied the hydrogen overflow in platinum @ MOF-801 under dry and wet hydrogen conditions. The authors claim that they have found a new water-assisted hydrogen spillover pathway with a lower migration barrier. However, this statement is not true, because in the field of heterogeneous catalysis, H₂O and other proton solvents have been promoting hydrogen spillover for more than 50 years. In the early 1970s, M. Boudart first reported that protonated solvents can help the migration of surface H in the form of H⁺/e pairs through solvation (hydrogen bonding). Therefore, the author's conclusion is not new.

I agree that so far, there has been no systematic study of hydrogen spillage on MOF-based materials. The current research is a good contribution, but there is a major logical defect. The MOF-801 structure collapsed in the hydrogen overflow experiment. At present, it is not clear whether the hydrogenation kinetics of ligands is limited by hydrogen spillover or MOF structure collapse/ligand diffusion rate. The co-flow of water not only promotes the overflow of hydrogen, but also promotes the structural collapse of metal oxides and the diffusion of ligands on the surface of metal catalysts. Therefore, the MOF-801 system is not an ideal model system for strictly studying the dynamics of hydrogen spillage. The research of catalytic reaction is also carried out under the condition of low scientific rigor. In general, I think this research cannot meet the high standard of Nature Communication and should be submitted to other journals specializing in material chemistry.

Reply: We thank the reviewer very much for the constructive comments. The hydrogen spillover researches on diverse catalytic carriers, such as TiO₂^{1, 2}, Cu³, zeolite⁴, MOF⁵, aluminosilicate matrix⁶, CNT⁷ and mesoporous organosilicas⁸ have received tremendous attentions and relevant work has been published in well-known journals, probably due to

following advantages. Firstly, hydrogen spillover can increase the content and distribution of activated hydrogen on the carrier, which helps to regulate the hydrogenation equilibrium with improved catalytic activity. Secondly, the new type catalytic sites can be created on carriers by activated hydrogen diffusion, which is beneficial to selectivity regulation. Thirdly, hydrogenation from hydrogen spillover can isolate noble metal centers from reactants, which not only prevents the noble metal center from poisoning, but also avoids excessive hydrogenation. Therefore, the research on hydrogen spillover, especially on the mechanism of and ways to enhance hydrogen spillover, can have significant implications and prospects.

In fact, we fully agree with the reviewer's viewpoint about "**H₂O and other proton solvents have been promoting hydrogen spillover**". As mentioned, the promotion of hydrogen spillover by water molecules has been reported on diverse substrate materials, such as metal oxides (Al₂O₃⁹, Fe₃O₄¹⁰, TiO₂²). We also discussed the effect of water on hydrogen spillover in the third paragraph of our manuscript and provided some references from Ref. 39-42. These researches reported diverse promotion mechanisms on different catalytic substrates. For instance, Lindsay R. Merte reported that H atoms could be combined with O of FeO to form OH group and be diffused by direct position transfer of a H₂O molecule from one surface OH group to construct another OH group¹¹. However, in MOFs, the distance between two metal clusters (which is determined by the ligand size in MOFs) is farther than the distance between two Fe atoms in FeO, which will hinder the direct diffusion of hydrogen atoms between different MOF clusters through the movement of a single water molecule (the mechanism on iron oxide). Moreover, the water molecules absorbed in MOFs are constrained by the water absorption sites and water absorption mechanisms, which restricts the free movement of absorbed water molecules within MOFs¹². Therefore, the promotion mechanism of hydrogen spillover by water molecule in FeO hardly explains the promotion mechanism in MOFs, which is worthy of further exploration.

Furthermore, MOFs as a type of well-known carrier material have attracted great interest in heterogeneous catalysis. Due to their superior features including large surface area, tunable pore structure, adjustable chemical composition, convenient surface functionalization, and well-defined metal sites, MOFs exhibited good catalytic performances in many fields such as thermal catalysis, photocatalysis and electrocatalysis, which are generally implemented by metal clusters, functional organic linkers or the guest materials¹³. By combining with noble metal nanomaterials, MOFs can regulate hydrogenation by adjusting electronic properties,

controlling size selectivity and concentrating gaseous reactants. Therefore, MOFs are a type of excellent hydrogenation carrier, implicating their potential application and research value for hydrogen spillover¹⁴. However, due to the high migration energy barrier of hydrogen atoms on the ligands, many researches have indicated that hydrogen spillover is limited in MOF carrier¹⁵. In fact, the existence of hydrogen spillover in MOFs had been in suspense for a long time, until the direct evidence of hydrogen spillover in ZIF-8 by Hua Chun Zeng⁵. Currently, the hydrogen spillover in MOFs is still faced with significant challenges including slow hydrogen atom mobility rate and short spillover distance in MOFs, poor thermal and chemical stability of most MOFs structures and complicated MOF microenvironment. Therefore, developing suitable strategy to improve hydrogen spillover in MOFs is meaningful for further advancement of MOF-based catalysts.

Fig. R1. WO₃ and Pt@110 nm MOF-801 in wet or dry H₂ at 200 °C for different lengths of time. (Revised Supplementary Figure 29)

Fig. R2. NMR curves of Pt@110 nm MOF-801(dry) for 2 h and Pt@110 nm MOF-801(wet) for 20 min. (Revised Supplementary Figure 30)

Fig. R3. PXRD patterns of Pt@110 nm MOF-801, Pt@110 nm MOF-801(dry) for 2 h and Pt@110 nm MOF-801(wet) for 20 min. (Revised Supplementary Figure 31)

Fig. R4. Nitrogen adsorption–desorption isotherms of Pt@110 nm MOF-801, Pt@110 nm MOF-801(dry) for 2 h and Pt@110 nm MOF-801(wet) for 20 min. (Revised Supplementary Figure 32)

Fig. R5. Pore size distribution of Pt@110 nm MOF-801, Pt@110 nm MOF-801(dry) for 2 h and Pt@110 nm MOF-801(wet) for 20 min. (Revised Supplementary Figure 33)

The reviewer commented that **“The MOF-801 structure collapsed in the hydrogen overflow experiment. At present, it is not clear whether the hydrogenation kinetics of ligands is limited by hydrogen spillover or MOF structure collapse/ligand diffusion rate.”** In our view, hydrogen spillover is the source of ligand hydrogenation and occurs much earlier than the ligand hydrogenation and MOF structure collapse. According to the literature¹¹, hydrogen spillover would reach equilibrium in a short period of time, and then H diffusion process completion was observed in a few seconds by scanning tunneling microscope (STM). Meanwhile, the collision theory of chemical reactions indicated that during the H diffusion, the ligand would undergo multitudinous collisions of hydrogen atoms for hydrogenation, so that the hydrogenation in MOFs require longer time which is affected by energy barriers and probabilities. At the same time, the collapse of MOF structure is caused by the ligand hydrogenation and occurs later than hydrogen spillover. Therefore, at the initial stage of the reaction, the hydrogen spillover in MOF-801 would not be limited by MOF structure collapse due to the earlier occurrence of hydrogen spillover. In order to demonstrate the concept, we studied the sequence of hydrogen spillover in MOFs and MOF structure collapse. In the case of Pt@110 nm MOF-801 in wet H₂, we observed slight discoloration of tungsten oxide after 5 min of thermal treatment (Fig. R1). This discoloration became significant after 20 minutes, despite only 7.6% ligand conversion (Fig. R2). These results confirmed that water-assisted hydrogen spillover occurs earlier than the collapse of MOF-801 structure. The XRD pattern (Fig. R3) and nitrogen adsorption–desorption isotherms (Fig. R4, 5) also supported these findings. By contrast, when dry H₂ was used in the thermal treatment of Pt@110 nm MOF-801, there was hardly any discoloration of tungsten oxide even after 2 h. This suggested that with the assistance of water molecules, hydrogen spillover seems to cover the region completely before structural damage takes place. Based on the above, the MOF-801 system as the research model for studying the dynamics of hydrogen spillover is not considered as a logical defect.

Moreover, in our strategy, the MOF-801 was selected as the research model by ligand conversion to intuitively demonstrate of hydrogen spillover in MOFs with at least four advantages. Firstly, the hydrogenation is easier to characterize than the signal of hydrogen spillover. Secondly, the hydrogenation of C=C bond can directly evidence the H diffusion on the ligand, which is beneficial for path identification of hydrogen spillover in MOFs. Thirdly, our research on the water absorption capacity and mechanism of MOF-801 is relatively clear and adequate, providing a theoretical basis and model for investigating water-assisted hydrogen

spillover. Lastly, compared to other signal conversions (such as tungsten oxide discoloration, oxide reduction or hydrogenolysis), hydrogenation can better demonstrate the hydrogenation ability and potential application of hydrogen spillover for catalysis.

As a well-known water-absorbing MOFs material, MOF-801 has been certified by many researchers in its water absorption stability. For instance, Omar M. Yaghi considered that MOF-801 exhibits the best and closest to the ideal cycling performance among several MOFs¹⁶ and shows robust cycling performance in the water uptake for all five cycles due to the no strong binding sites (e.g., open metal sites) for water to bind¹⁷. And the effects of water, hydrogen, and Pt nanoparticles on the structural stability of MOF-801 are provided by control experiments in the manuscript. From the experimental results, it was found that water or hydrogen exposes hardly impact on the crystal structure of MOF-801 at the treatment temperature. In other words, MOF-801 will not collapse in structure within 200 °C. The humidity of wet H₂ at 200 °C is around 0.19% RH (Methods of Manuscript), which is fairly low for stability damage of MOF-801. In a word, MOF-801 is stable in our research environment at 200 °C.

The poisoning of catalysts in catalytic reaction system is indeed a common occurrence that poses a challenge in the field of catalysis. Maintaining the activity of catalysts while avoiding poisoning is a hot topic of research. One approach to address this issue is by encapsulating active centers in MOF materials, where the pore size selectivity of MOFs can control the entry of reactants and the resistance of toxicant. However, such selectivity would be limited when the reactants and toxicant have similar sizes. Meanwhile, by introducing hydrogenation ability to the surface of MOF crystal via hydrogen spillover, MOFs with small-sized window could hydrogenate large-sized reactants with large-sized poison being blocked, which ensures the operation of catalytic reactions in complex catalytic environments through hydrogen spillover. In addition, hydrogen spillover can create new active sites for hydrogenation, potentially altering the molecular activation process and reducing the energy barrier for reactants³. The elements in microenvironment of MOFs, including adjustable clusters, ligands and guest molecules, possess the potential as unique active sites to participate in hydrogenation through hydrogen spillover. Therefore, the proposed water-assisted hydrogen spillover strategy provides more possibilities for the catalytic application of MOF-based catalyst.

Discussions on the impact of structure collapse on hydrogen spillover have been added to the eighth paragraph in the Manuscript. And the relevant figures have been added to Supplementary Figure 29-33 in the Supporting Information.

“In the process of water-assisted hydrogen spillover in MOF-801, the ligand hydrogenation and MOF structure collapse were observed, which raised questions about their influence on hydrogen spillover. During the hydrogen spillover process in MOF-801, hydrogen spillover is considered as the source of ligand hydrogenation and occurs much earlier than the ligand hydrogenation and MOF structure collapse. Studies using scanning tunneling microscope (STM) indicated that hydrogen spillover would reach equilibrium in a short period of time, typically completing H diffusion process in a few seconds¹¹. Meanwhile, the collision theory of chemical reactions indicated that during the H diffusion, the ligand would undergo multitudinous collisions of hydrogen atoms for hydrogenation, so that the hydrogenation in MOFs required longer time which was affected by energy barriers and probabilities. At the same time, the collapse of MOF structure was caused by the ligand hydrogenation and occurred later than hydrogen spillover. Therefore, at the initial stage of the reaction, the hydrogen spillover in MOF-801 would not be limited by MOF structure collapse due to the earlier occurrence of hydrogen spillover. In order to demonstrate the concept, we studied the sequence of hydrogen spillover in MOFs and MOF structure collapse. In the case of Pt@110 nm MOF-801 in wet H₂, we observed slight discoloration of tungsten oxide after 5 min of thermal treatment. This discoloration became significant after 20 min, despite only 7.6% ligand conversion. These results confirmed that water-assisted hydrogen spillover occurs earlier than the collapse of MOF-801 structure. The XRD pattern and nitrogen adsorption–desorption isotherms also supported these findings. By contrast, when dry H₂ was used in the thermal treatment of Pt@110 nm MOF-801, there was hardly any discoloration of tungsten oxide even after 2 h. This suggested that with the assistance of water molecules, hydrogen spillover seems to cover the region completely before structural damage takes place. Therefore, the hydrogen spillover region can be estimated by analyzing the ligand hydrogenation.”

Discussions on the advantages of MOF-801 as hydrogen spillover model have been added to the fourth paragraph.

“In our strategy, MOF-801 was selected as the research model by ligand conversion to intuitively demonstrate hydrogen spillover in MOFs with at least four advantages. Firstly, the hydrogenation is easier to characterize than the signal of hydrogen spillover. Secondly, the hydrogenation of C=C bond can directly evidence the H diffusion on the ligand, which is beneficial for path identification of hydrogen spillover in MOFs. Thirdly, our research on the water absorption capacity and mechanism of MOF-801 is relatively clear and adequate, providing a theoretical basis and model for investigating water-assisted hydrogen spillover. Lastly, compared with other signal conversions (such as tungsten oxide discoloration, oxide reduction or hydrogenolysis), hydrogenation can better demonstrate the hydrogenation ability and potential application of hydrogen spillover for catalysis. Here, MOF-801 and Pt@MOF-801 were synthesized respectively by the reported literature¹⁸ and the controlled nanoparticle encapsulation strategy¹⁹.”

Reviewer 2:

In this manuscript, Zhida Gu et al report on water-assisted hydrogen spillover in metal–organic frameworks. Hydrogen spillover phenomenon was detected by ligand conversion and structural changes as detectors in MOF-801. The mechanism was also discussed by deuterium-labeled experiments or DFT calculation. The enhanced hydrogen spillover by water can be applied to more MOFs or COFs materials. The results in this manuscript are surely interesting. However, there are several issues that the authors need to correspond. Once addressed the issues below, I like to reconsider the manuscript to be suitable for publication in Nature Communications.

Reply: We thank the reviewer very much for the positive and valuable comments. We have revised the Manuscript carefully according to the suggestions.

Q1. In Fig. 2g, the structure of Pt@MOF-801 dramatically changed by wet H₂ thermal treatment at 30 °C. However, as shown in Fig. 2h, hydrogen consumption is negligible up to 250 °C. The authors are required to explain the reason.

Reply: Many thanks to the reviewer for raising the question. In our manuscript, most thermal treatment occurs at 200 °C. And all samples under dry or wet H₂ in Fig. 2g are also treated at 200 °C.

Temperature programmed reduction (TPR) characterization typically requires an anhydrous environment. This means that TPR tests for all the samples are carried out in dry H₂ condition.

Therefore, the TPR result that shows no hydrogen consumption of the samples up to 250 °C is consistent with the observation of no obvious change in XRD patterns for the samples during dry H₂ thermal treatment.

In the fifth paragraph, the temperature of thermal treatment is added and marked.

“To investigate efficiency enhancement of hydrogen spillover by assistance of water molecules in MOF-801, samples including Pt@150 nm MOF-801, Pt@110 nm MOF-801 and the mixtures of Pt NPs with MOF-801 (named as Pt-MOF-801) were respectively treated at 200 °C for 2 h in input H₂ under two humidity conditions: dry H₂ (28 °C and 20% RH) and wet H₂ (28 °C and 85% RH).”

Q2. In addition, in Fig. 2g, hydrogen consumption of Pt@MOF-801 is larger than that of Pt@MOF-801(wet). Why?

Reply: Many thanks for the question. When conducting TPR in an anhydrous condition, achieving in-situ hydrogen consumption under wet H₂ treatment can be challenging. Hence, Pt@MOF-801(wet), which has been thermally treated in wet H₂ at 200 °C for 2 h, witnessed a considerable proportion of ligands to undergo hydrogenation due to the occurrence of hydrogen spillover. In the TPR measurement, Pt@MOF-801(wet) exhibited a lower hydrogen consumption than Pt@MOF-801 mainly due to hydrogen pre-consumption in the wet H₂ treatment by hydrogen spillover.

In the fifth paragraph, the explanation of thermal treatment is modified and marked.

“The Pt@150 nm MOF-801(wet), Pt@110 nm MOF-801(wet) and Pt-MOF-801(wet) initially showed similar tendency at 300 °C but obvious lower intensity of hydrogen consumption at 400 °C than themselves without wet H₂ treatment. This intensity difference in hydrogen consumption can be attributed to hydrogen pre-consumption during the wet H₂ treatment of samples by hydrogen spillover.”

Q3. In Fig. 10S, the hydrogenation reaction of Pt@MOF-801(dry) seems to be progressing more than Pt@MOF-801(wet).

Reply: Many thanks for the correction. We did make the mistake to put the tag in the wrong position in original Supplementary Figure 10, which has now been corrected and shown as Fig.R6.

The figure has been corrected as the Supplementary Figure 21 in supporting information.

Fig. R6. NMR curves of Pt-FA(dry), Pt@150 nm MOF-801(dry), Pt@150 nm MOF-801(wet), Pt@110 nm MOF-801(dry), Pt@110 nm MOF-801(wet), Pt-MOF-801(dry), and Pt-MOF-801(wet). (Revised Supplementary Figure 21)

Q4. In Fig.7, the catalytic tests of 20% Pt/C should be performed under the same condition of Pt@ZIF-8 for appropriate comparison.

Reply: Thanks for the constructive suggestion. For 2,6-dimethylnitrobenzene hydrogenation, Pt@ZIF-8 exhibited no activity in ethyl acetate at 30 °C for 12 h, 2.27% conversion in dry H₂ at 200 °C for 8 h and 39.26% conversion in wet H₂ at 200 °C for 8 h (Fig. R7), which proved both the inaccessibility of macromolecular reactants with Pt NPs in MOFs and the catalytic activity improvement due to the hydrogen spillover enhancement using our strategy. The 20% Pt/C catalyst was tested at 30°C and 200 °C with 69.57% and 96.51% conversion respectively. When quinoline was introduced as the toxicant, both Pt@ZIF-8 and the 20% Pt/C catalyst at 200 °C experienced slight decreases in conversion (34.8% and 92.01%, respectively). This indicates a limited poisoning effect, which can be attributed to the easy desorption and gasification of quinoline molecules at high temperatures, rendering the quinoline toxicant ineffective. However, these results alone may not be sufficient to prove the antitoxicity of Pt@ZIF-8. To further evaluate the antitoxicity, another toxicant, 4,4'-dithiodipyridine (DTDP), was selected as toxicant model molecule with stronger adsorption (two pyridine rings and two sulfur atoms in one DTDP molecule) and higher boiling point (356.1 °C). With DTDP as the toxicant, the 20% Pt/C catalyst exhibited 1.53% and 11.86% conversion at 30 °C and 200 °C respectively, indicating a clear poisoning effect of DTPP with a reduction in catalytic activity (Fig. R8). In contrast, Pt@ZIF-8 maintained a conversion (35.60%) with DTDP as the toxicant in wet H₂ at 200 °C, demonstrating the conversion improvement and antitoxicity achieved through water-assisted hydrogen spillover strategy.

Fig. R7. Nitro hydrogenation conversion including commercial 20% Pt/C, commercial 20% Pt/C with toxicant, Pt@ZIF-8 at 30 °C. And the nitro hydrogenation conversion including Pt@ZIF-8 in dry H₂, Pt@ZIF-8 in wet H₂, Pt@ZIF-8 with quinoline in wet H₂, commercial 20% Pt/C and commercial 20% Pt/C with quinoline at 200 °C.

Fig. R8. Nitro hydrogenation conversion including commercial 20% Pt/C, commercial 20% Pt/C with toxicant, Pt@ZIF-8 at 30 °C. And the nitro hydrogenation conversion including Pt@ZIF-8 in dry H₂, Pt@ZIF-8 in wet H₂, Pt@ZIF-8 with DTDP in wet H₂, commercial 20% Pt/C and commercial 20% Pt/C with DTDP at 200 °C.

Part of the descriptions in the penultimate paragraph and Fig. 7 of the Manuscript have been modified and marked as Fig. R9.

*“Herein, the 2,6-dimethylnitrobenzene (4.02 *8.32 *8.9 Å) and 4,4'-dithiodipyridine (DTDP) (5.30 *6.46 *12.88 Å) with larger molecule size than the window of ZIF-8 (3.4 Å) were selected as model reactant for nitro hydrogenation, and model toxicant due to the high toxicity of sulfur and pyridine compounds to noble metal centers (Fig. 7a). Owing to the steric effect, Pt@ZIF-8 exhibited no activity on 2,6-dimethylnitrobenzene hydrogenation in ethyl acetate at 30 °C for 12 h, which proved that the macromolecular reactants are inaccessible to Pt NPs in MOFs. Meanwhile, the commercial 20% Pt/C catalyst had an appreciable conversion (69.57%) for 2,6-dimethylaniline under similar conditions; however, it would decrease to 1.53% after being poisoned by the introduction of DTDP. Furthermore, in order to hydrogenate macromolecular reactant at the outside of MOFs by hydrogen spillover, the hydrogenation was investigated in dry or wet H₂ conditions at 200 °C without additional solvent for 8 h. The Pt@ZIF-8 had a low conversion (2.27%) in dry H₂, which is attributed to the low efficiency of hydrogen spillover in ZIF-8. On the contrary, the conversion in wet H₂ had an obvious improvement (39.26%) due to the efficiency enhancement of hydrogen spillover by our strategy. More importantly, the Pt@ZIF-8 has still showed the similar conversion (35.60%) with the existence of toxicant, mainly due to the protection of MOF and the improved hydrogen spillover. Meanwhile, the commercial 20% Pt/C catalyst showed a significant drop in conversion from 96.51% to 11.86%*

at 200 °C. We believed that the conversion improvement and antitoxicity based on the design of hydrogen spillover in MOFs represent a promising beginning and highlight the potential of this approach in catalytic research.”

Fig. R9. Antitoxic nitro hydrogenation at the outside of Pt@ZIF-8 by water-assisted hydrogen spillover. (a) Scheme of nitro hydrogenation by Pt@ZIF-8 under dry or wet H₂ environments. (b) Nitro hydrogenation conversion at 30 °C including commercial 20% Pt/C, commercial 20% Pt/C with DTDP and Pt@ZIF-8, respectively. And nitro hydrogenation conversion at 200 °C including Pt@ZIF-8 in dry H₂, Pt@ZIF-8 in wet H₂, Pt@ZIF-8 under DTDP in wet H₂, commercial 20% Pt/C and commercial 20% Pt/C under DTDP, respectively. (Revised Fig. 7)

Part of catalytic method have been modified and marked in Methods of Manuscript.

“Catalysis of 2,6-dimethylnitrobenzene hydrogenation

For 2,6-dimethylnitrobenzene hydrogenation at 30 °C, 20 mg Pt@ZIF-8 or 10 mg commercial 20% Pt/C catalyst was dispersed in 0.5 mL 2,6-dimethylnitrobenzene and 3 mL ethyl acetate. And then 5 mg DTDP was provided additionally to investigate catalyst antitoxicity. The hydrogenation took place in a closed high-pressure reactor with 60 mL volume and initial 0.4 MPa hydrogen at 30 °C for 12 h. The reaction products were directly monitored by GC.

For 2,6-dimethylnitrobenzene hydrogenation at 200 °C, 20 mg Pt@ZIF-8 was degassed at 180 °C for 12 h under vacuum, and dispersed in 0.5 mL 2,6-dimethylnitrobenzene. Additionally, 0.2 mL water was provided for humidity increase, and 5 mg DTDP was provided for investigating catalyst antitoxicity. Through inductive coupled plasma emission testing, Pt ratio in Pt@ZIF-8 was calculated to be around 1.45%. And the Pt content in 20 mg Pt@ZIF-8 was equivalent to that in ~1.5 mg commercial 20% Pt/C catalyst. To compare catalyst antitoxicity, 1.5 mg commercial 20% Pt/C catalyst was dispersed in 0.5 mL 2,6-dimethylnitrobenzene with and without additional 5 mg DTDP at 200 °C, respectively. The

hydrogenation was reacted in a closed high-pressure reactor with 60 mL volume and initial 0.4 MPa hydrogen at 200 °C for 8 h. The reaction products were directly monitored by GC.”

Q5. The authors only showed the color change before/after H₂ treatment to indicate efficiency promotion of hydrogen spillover of other MOFs or covalent organic framework materials (Fig. 6), but it is insufficient. The characterization of their materials after H₂ treatment is required eg XRD, N₂ sorption.

Reply: Thank you for the suggestions. Here, MOF and COF structures after H₂ treatment were characterized by XRD and nitrogen adsorption–desorption. After dry or wet hydrogen thermal treatment, Pt@UiO-66, Pt@UiO-67, Pt@ZIF-8 and Pt@ ZIF-67 maintained the similar XRD patterns (Fig. R10), nitrogen adsorption–desorption curves and pore size distributions (Fig. R11). These data indicated the negligible influence of hydrogen spillover on their structure, which reflected that UiO-66, UiO-67, ZIF-8 and ZIF-67 were relatively stable for hydrogen spillover. Furthermore, Pt@HKUST-1, Pt@Fe-MIL-53, Pt/TAPT-DHTA and Pt/TpPa-1 exhibited crystal structure damage (Fig. R10) and declined BET surface area (Fig. R12) after dry or wet hydrogen thermal treatment. On the one hand, according to the obvious influence of hydrogen spillover on their structure, Pt@HKUST-1, Pt@Fe-MIL-53, Pt/TAPT-DHTA and Pt/TpPa-1 were relatively unstable for hydrogen spillover, which may be caused by the reducible metal clusters or hydrogenable ligands. Interestingly, the damage to Pt@HKUST-1, Pt@Fe-MIL-53, and Pt/TAPT-DHTA was more pronounced after wet hydrogen thermal treatment compared to dry hydrogen thermal treatment.

Fig. R10. PXRD patterns of (a) Pt@UiO-66, (b) Pt@UiO-67, (c) Pt@ZIF-8, (d) Pt@ZIF-67, (e) Pt@HKUST-1, (f) Pt@Fe-MIL-53, (g) Pt/TAPT-DHTA and (h) Pt/TpPa-1 after dry or wet H_2 treatment at 200 °C. (Revised Supplementary Figure 39)

Fig. R11. Nitrogen adsorption–desorption isotherms and pore size distribution of (a) Pt@UiO-66, (b) Pt@UiO-67, (c) Pt@ZIF-8 and (d) Pt@ZIF-67 after dry or wet H₂ treatment at 200 °C. (Revised Supplementary Figure 40)

Fig. R12. Nitrogen adsorption–desorption isotherms and pore size distribution of (a) Pt@HKUST-1 (b) Pt@Fe-MIL-53, (c) Pt/TAPT-DHTA and (d) Pt@TpPa-1 after dry or wet H₂ treatment at 200 °C. (Revised Supplementary Figure 41)

Part of descriptions from the third to the last paragraphs in the Manuscript and Supplementary Figure 39-41 in the Supporting Information have been modified and marked.

“After dry or wet hydrogen thermal treatment, Pt@UiO-66, Pt@UiO-67, Pt@ZIF-8 and Pt@ZIF-67 maintained similar XRD patterns (Supplementary Fig. 39), nitrogen adsorption–desorption curves and pore size distributions (Supplementary Fig. 40). These data indicated the negligible influence of hydrogen spillover on their structure, which reflected that UiO-66, UiO-67, ZIF-8 and ZIF-67 are relatively stable for hydrogen spillover. Furthermore, Pt@HKUST-1, Pt@Fe-MIL-53, Pt/TAPT-DHTA and Pt/TpPa-1 exhibited crystal structure damage (Supplementary Fig. 39) and declined BET surface area (Supplementary Fig. 41) after dry or wet hydrogen thermal treatment. On the one hand, according to the obvious influence of hydrogen spillover on their structure, Pt@HKUST-1, Pt@Fe-MIL-53, Pt/TAPT-DHTA and Pt/TpPa-1 were relatively unstable for hydrogen spillover, which may be caused by the reducible metal clusters or hydrogenable ligands. Interestingly, the damage to Pt@HKUST-1, Pt@Fe-MIL-53, and Pt/TAPT-DHTA was more pronounced after wet hydrogen thermal treatment compared to dry hydrogen thermal treatment. Such universal strategy of water-assisted hydrogen spillover can be extended not only to stable MOFs for the design of porous crystalline catalysts, but also to unstable porous materials for the preparation of derivatives.”

Reviewer 3:

Hydrogen spillover is a very important interfacial phenomenon that has been well-documented in hydrogenation reactions. And, there have been many studies to report the existence of hydrogen spillover effect in noble metal-encapsulated MOFs or COFs. However, this research field is still in its infancy and more efforts are needed to explore more effective detection methods of this phenomenon and to make clear its reaction mechanism. This study prepared a Pt nanoparticles@MOF-801 material to try to address some of the existing problems. In view of the literature, this work is not much new in terms of material preparation and subsequent mechanism verification. It is important that the authors chose a good MOF host where its ligands with the C=C bond could serve a built-in detector for those activated hydrogen atoms by Pt, and that they reported a possibly new hydrogen spillover path assisted by water molecules adsorbing within this MOF host. However, I find that there are many important issues that needed to be addressed carefully, as follows:

Reply: We thank the reviewer very much for the positive and valuable comments. We have revised the Manuscript carefully according to the suggestions.

Q1. The title is a little misleading and exaggerated, which ignored the key function of Pt.

Reply: Thank you for the suggestions. The research focus in our manuscript is the capacity, mechanism, extensibility and application of hydrogen spillover on MOFs. Indeed, Pt nanoparticles play a role in generating H atoms, which is important role in hydrogen spillover. Therefore, we modified the title to “**Water-assisted hydrogen spillover in Pt nanoparticle-based metal–organic framework composites**”.

Q2. The hydrogen migration distance should be expressed in terms of radius (namely shell thickness) rather than diameter centered on the location of these Pt atoms. It is misleading here. In this regard, Ref. 23 obviously reported a much larger region than this study, rather than “from imperceptible existence” the authors said here.

Reply: Thank you for the suggestions. Here, both the shell thickness mentioned in Ref. 23 and the hydrogen spillover region in our work are different yet common ways of describing the influence of hydrogen spillover. For example, when describing the power of a bomb (or nuclear bomb), the explosion diameter (radius) is often used as a measure, which is consistent with our concept of hydrogen spillover region. In particular, the expression of hydrogen spillover region here holds practical significance in determining whether hydrogen atoms can diffuse beyond the catalyst structure of Pt@MOFs.

Ref. 23 presented remarkable research in the field of hydrogen spillover in MOFs. However, the reported hydrogen spillover region in Ref. 23 exhibited 10-20 nm at 220-240 °C, which was comparatively smaller than that observed in some other materials. Furthermore, the hydrogen spillover region in our research cannot be directly compared to Ref. 23. Firstly, the temperatures of hydrogen spillover in Ref. 23 are 220 °C, 240 °C and 260 °C, whereas our research focuses on a temperature of 200 °C. As the region of hydrogen spillover is related to temperature, direct comparisons of hydrogen spillover between different temperature ranges are not appropriate. Secondly, ZIF-8 and MOF-801 have distinct metal clusters, ligands, coordination modes, and pore structures, which can lead to different hydrogen spillover capabilities in different MOFs. Therefore, the conclusions drawn regarding the hydrogen spillover capability in ZIF-8 cannot be directly applied to MOF-801. Lastly, the disintegration in Ref. 23 and ligand hydrogenation in our manuscript represent different evaluation indicators of hydrogen spillover, with differing energy barriers for the respective reactions. Based on these considerations, the use of different evaluation indicators can lead to variations in the conclusions drawn regarding hydrogen spillover capabilities. Moreover, the indicators of ligand hydrogenation in our manuscript could provide potential assistance for the application of hydrogen spillover in catalysis. Importantly, our manuscript has a distinct focus from Ref. 23. While Ref. 23 concentrates on the hydrogen spillover in ZIF-8 under different conditions,

our manuscript aims to develop a strategy for enhancing hydrogen spillover in diverse MOFs, including ZIF-8. Consequently, both our manuscript and Ref. 23 represent excellent works in investigating hydrogen spillover in MOFs, albeit with different research topics and focuses.

Q3. The samples' names in many Supplementary Figures are very chaotic. It is also difficult to distinguish “D” and “O” in Fig. 4. Some of the key data occluded each other. In addition, the authors are suggested to name (and rename) the samples clearly at the very beginning. MOF-81 should be clearly introduced, including its compositions, structure, the position and amount of its adsorbed water, etc. All in all, the present manuscript is hard to read and the authors should carefully refine and rearrange some of the contents. By the way, there are also some errors, like at Page 6 “Whist under wet H₂ treatment...”, Supplementary Figure 4 (the temperature should be 300 °C rather than 200 °C they discussed in the test “MOF-801 were tested respectively in wet H₂ and wet N₂ conditions at 200 °C”), and Fig. 1 (which can't be found in the test).

Reply: Thank you very much for your valuable suggestions. We have corrected the errors in the Manuscript piece by piece. Firstly, the introduction to MOF-801 has been added in the Manuscript. Secondly, at the beginning of characterization, the samples after thermal treatment were also named. Thirdly, Fig. 4 in the Manuscript has been rearranged and corrected as Fig. R13. Fourthly, the Manuscript has been carefully polished and corrected. Fifthly, the original Supplementary Figure 4, 5, 6 have been divided into several figures in the Supporting Information for better reading. Lastly, Fig. 1 has been marked in the Manuscript. And thank you again for correcting the errors in this Manuscript.

The introduction to MOF-801 has been added in the fourth paragraph.

“MOF-801 is a type of MOF with an fcu topology. It consists of $Zr_6O_4(OH)_4(-CO_2)_n$ as clusters and fumaric acid as the ligands, which was first reported by Peter Behrens et al²⁰. Due to its similar topological structure to UiO-66, MOF-801 demonstrates excellent thermal stability (260 °C in air), superior water absorption capacity at low humidity (2.8 liters of water per kilogram of MOF per day at 20% relative humidity (RH))¹⁶, and water stability for at least a week²⁰.”

Names of samples have been added in the fifth paragraph.

“In addition, MOF-801 exhibited outstanding cycling performance and showed reliable stability in the water uptake for all five cycles¹⁷. Furthermore, more control experiments on the stability of MOF-801 were investigated including the effect of H₂, humidity and Pt NPs. Specifically, MOF-801 in wet H₂ (28 °C and 85% RH) thermal treatment at 200 °C (Supplementary Fig. 9a) and Pt@150 nm MOF-801 in wet N₂ (28 °C and 80% RH) thermal treatment at 200 °C (Supplementary Fig. 9b) were tested respectively.”

Fig. 4 in the Manuscript has been rearranged and corrected as Fig. R13.

Fig. R13. Deuterium labeling experiments for characterizing the effect of water in water-assisted hydrogen spillover. The ligand conversion of MOF-801 after thermal treatment in H₂O-containing (a) D₂ and (b) D₂O-containing H₂. The ligand conversion of Pt@110 nm MOF-801 after thermal treatment in (c) H₂O-containing D₂ and (d) D₂O-containing H₂. (Revised Fig. 4)

The original Supplementary Figure 4, 5, 6 have been divided into Fig. R14-22 for better

reading.

Fig. R14. PXRD patterns of MOF-801 simulation, MOF-801, Pt@150 nm MOF-801 and Pt@110 nm MOF-801. (Revised Supplementary Figure 5)

Fig. R15. Nitrogen adsorption–desorption isotherms of MOF-801, Pt@150 nm MOF-801 and Pt@110 nm MOF-801. (Revised Supplementary Figure 6)

Fig. R16. Pore size distribution of MOF-801, Pt@150 nm MOF-801 and Pt@110 nm MOF-801. (Revised Supplementary Figure 7)

Fig. R17. PXRD patterns of MOF-801 in wet H₂ at 200 °C and Pt@150 nm MOF-801 in wet N₂ at 200 °C. (Revised Supplementary Figure 10)

Fig. R18. Nitrogen adsorption–desorption isotherms of MOF-801, MOF-801 in wet H₂ at 200 °C and Pt@150 nm MOF-801 in wet N₂ at 200 °C. (Revised Supplementary Figure 11)

Fig. R19. Pore size distribution of MOF-801, MOF-801 in wet H₂ at 200 °C and Pt@150 nm MOF-801 in wet N₂ at 200 °C. (Revised Supplementary Figure 12)

Fig. R20. PXRD patterns of Pt@150 nm MOF-801 and Pt@150 nm MOF-801 in H₂ at 300 °C. (Revised Supplementary Figure 18)

Fig. R21. Nitrogen adsorption–desorption isotherms of Pt@150 nm MOF-801 and Pt@150 nm MOF-801 in H₂ at 300 °C. (Revised Supplementary Figure 19)

Fig. R22. Pore size distribution of Pt@150 nm MOF-801 and Pt@150 nm MOF-801 in H₂ at 300 °C. (Revised Supplementary Figure 20)

Fig. 1 has been marked in the third paragraph of Manuscript.

“Specifically, Pt@MOF-801 was designed by ingeniously encapsulating Pt nanoparticles (Pt NPs) in the center of MOF-801 in which Pt NPs served as the hydrogen dissociation sites while the C=C in the MOF ligand as the acceptor of activated hydrogen atom (Fig. 1).”

Q4. More clear TEM characterization results about Pt nanoparticles should be provided. And what are their sizes and weight ratios in each sample? Would these influence the performance analysis? What is the action of Zr cluster? Where are the μ_3 -OH groups?

Reply: Thank you for the suggestions and questions. The size of Pt nanoparticles is around 4 nm (Fig. R23a). The weight ratios of Pt nanoparticles in Pt@110 nm MOF-801 and Pt@150 nm MOF-801 are 0.46% and 0.31% respectively. In fact, the size and weight ratios of Pt have an impact on the activation of hydrogen. To eliminate this effect, Pt@110 nm MOF-801 and Pt@150 nm MOF-801 were synthesized using identical methods, except that the synthetic time was 9 h and 18 h respectively. In this way, the two samples can have similar distribution of Pt nanoparticles, only with different diameters of MOF particles.

The zirconium-oxide clusters in MOF-801 play a crucial role in the hydrogen spillover process. Typically, the spillover path in MOFs is cluster-ligand-cluster. Metal-oxygen clusters are an indispensable part of this traditional hydrogen spillover path in MOFs with a lower spillover barrier than ligands²¹. In our research, we have developed a new water-assisted spillover path, which is cluster-water-cluster. The water molecules are adsorbed by the μ_3 -OH groups on the

zirconium-oxide clusters. Therefore, the zirconium-oxide clusters play roles in both adsorbing water molecules and serve as indispensable part of water-assisted hydrogen spillover.

The presence of μ_3 -OH groups on the zirconium-oxygen clusters has been reported in several literatures^{17,22}. Here, in order to provide a clearer position of μ_3 -OH in MOF-801, the schematic diagram of the μ_3 -OH position was provided in Fig. R23b.

Fig. R23. (a) TEM image of Pt nanoparticles, and (b) the μ_3 -OH on the zirconium-oxide cluster.

TEM image of Pt nanoparticles was added in Supplementary Figure 3d.

Fig. R24. TEM images of (a) MOF-801, (b) Pt@150 nm MOF-801, (c) Pt@110 nm MOF-801, (d) Pt nanoparticles. (Revised Supplementary Figure 3)

The water stability introduction of MOF-801 has been added in the fourth paragraph.

“In addition, MOF-801 exhibited outstanding cycling performance and showed reliable stability in the water uptake for all five cycles¹⁷. Furthermore, more control experiments on the stability of MOF-801 were investigated including the effect of H₂, humidity and Pt NPs. Specifically, MOF-801 in wet H₂ (28 °C and 85% RH) thermal treatment at 200 °C (Supplementary Fig. 9a) and Pt@150 nm MOF-801 in wet N₂ (28 °C and 80% RH) thermal treatment at 200 °C (Supplementary Fig. 9b) were tested respectively.”

The action of Zr cluster has been added in the ninth paragraph.

“As a crucial role in MOF-801, zirconium-oxide clusters are an indispensable part of the traditional hydrogen spillover path in MOFs, the diffusion of which is not the kinetical rate-limiting step with a lower spillover barrier than that of ligands in the spillover process²¹.”

Q6. Are the TEM images in Fig. 2d-f representative? More statistical images should be provided. And, to prove the reason for the adhesion phenomenon, the authors should prepare samples at a temperature below 188 °C.

Reply: Thank you for the suggestions. Here, more TEM images of Pt@150 nm MOF-801(wet), Pt@110 nm MOF-801(wet) and Pt-MOF-801(wet) (original Fig. 2d-f) were provided in Fig. R25. And we also prepare the Pt-MOF-801 at 160 °C (Fig. R26), which exhibited no adhesion phenomenon for not reaching the melting temperature.

Fig. R25. TEM images of (a) Pt@150 nm MOF-801(wet), (b) Pt@110 nm MOF-801(wet) and (c) Pt-MOF-801(wet). (Revised Supplementary Figure 15)

Fig. R26. (a) TEM image and (b) XRD pattern of Pt-MOF-801 in 160 °C wet H₂. (Revised Supplementary Figure 16)

In the fifth paragraph, the explanation of melted point is modified and marked.

“The melted amber acid might be the reason behind this adhesion on the surface of MOF nanoparticles, which could be supported by the absence of adhesion in wet H₂ at 160 °C (Supplementary Fig. 16).”

Q7. The structure images for various MOFs are too large in Fig. 7. The authors should clearly present their results rather than such information.

Reply: Thank you for the suggestions. Fig. 7 has been rearranged as Fig. R27.

Fig. R27. The structure information, TEM images and photographs of WO_3 and samples including UiO-66 (a), UiO-67 (b), ZIF-8 (c), ZIF-67 (d), HKUST-1 (e), Fe-MIL-53 (f), TAPT-DHTA (g) and TpPa-1 (h) after H_2 treatment in different humidities showing universality of the water-assisted hydrogen spillover strategy in various porous crystalline material. (Revised Fig. 7)

Q8. What is the evidence for the peak splitting of the XPS spectra? Reference should be added.

Reply: Thank you for your suggestion. Here, two references have been added for the splitting XPS spectra^{23, 24}.

Fig. R28. XPS spectrum of C 1s peaks of modified MWCNT²³ and MWCNTs²⁴.

The references were added in the sixth paragraph of Manuscript.

“To further support the results of spillover region, the X-ray photoelectron spectroscopy (XPS) was used for surface hydrogenation detection^{23, 24}.”

Thank you so much for your kind consideration, and we are looking forward to your response.

Sincerely yours,

Fengwei Huo

References:

1. Karim, W. et al. Catalyst support effects on hydrogen spillover. *Nature* **541**, 68–71 (2017).
2. Wei, J. et al. In situ Raman monitoring and manipulating of interfacial hydrogen spillover by precise fabrication of Au/TiO₂/Pt sandwich structures. *Angew. Chem. Int. Ed.* **59**, 10343–10347 (2020).
3. Jiang, L. et al. Facet engineering accelerates spillover hydrogenation on highly diluted metal nanocatalysts. *Nat. Nanotechnol.* **15**, 848–853 (2020).
4. Wang, S. et al. Activation and spillover of hydrogen on sub-1 nm palladium nanoclusters confined within sodalite zeolite for the semi-hydrogenation of alkynes. *Angew. Chem. Int. Ed.* **58**, 76681–7672 (2019).
5. Zhan, G. & Zeng, H.C. Hydrogen spillover through Matryoshka-type (ZIFs@)_{n-1}ZIFs nanocubes. *Nat. Commun.* **9**, 3778 (2018).
6. Im, J., Shin, H., Jang, H., Kim, H. & Choi, M. Maximizing the catalytic function of hydrogen spillover in platinum-encapsulated aluminosilicates with controlled nanostructures. *Nat. Commun.* **5**, 3370 (2014).
7. Briggs, N.M. et al. Identification of active sites on supported metal catalysts with carbon nanotube hydrogen highways. *Nat. Commun.* **9**, 3827 (2018).
8. Zou, H. et al. Dual metal nanoparticles within multicompartmentalized mesoporous organosilicas for efficient sequential hydrogenation. *Nat. Commun.* **12**, 4968 (2021).
9. Luo, J. & Epling, W.S. New insights into the promoting effect of H₂O on a model Pt/Ba/Al₂O₃ NSR catalyst. *Appl. Catal. B Environ.* **97**, 236–247 (2010).
10. Doudin, N. et al. Understanding heterolytic H₂ cleavage and water-assisted hydrogen spillover on Fe₃O₄(001)-supported single palladium atoms. *ACS Catal.* **9**, 7876–7887 (2019).
11. Merte, L.R. et al. Water-mediated proton hopping on an iron oxide surface. *Science* **336**, 8891–893 (2012).
12. Hanikel, N. et al. Evolution of water structures in metal–organic frameworks for improved atmospheric water harvesting. *Science* **374**, 454–459 (2021).
13. Li, G., Zhao, S., Zhang, Y. & Tang, Z. Metal–organic frameworks encapsulating active nanoparticles as emerging composites for catalysis: recent progress and perspectives. *Adv. Mater.* **30**, 1800702 (2018).
14. Yang, Q., Xu, Q. & Jiang, H. Metal–organic frameworks meet metal nanoparticles: synergistic effect for enhanced catalysis. *Chem. Soc. Rev.* **46**, 4774–4808 (2017).
15. Mavrandonakis, A. & Klopffer, W. Comment on “Kinetics and mechanistic model for hydrogen spillover on bridged metal–organic frameworks”. *J. Phys. Chem. C* **112**, 3152–3154 (2008).
16. Kim, H. et al. Water harvesting from air with metal–organic frameworks powered by natural sunlight. *Science* **356**, 430–434 (2017).
17. Furukawa, H. et al. Water adsorption in porous metal–organic frameworks and related materials. *J. Am. Chem. Soc.* **136**, 4369–4381 (2014).

18. Wang, X.G., Xu, L., Li, M.J. & Zhang, X.Z. Construction of flexible-on-rigid hybrid-phase metal–organic frameworks for controllable multi-drug delivery. *Angew. Chem. Int. Ed.* **59**, 18078–18086 (2020).
19. Zhang, W. et al. A family of metal–organic frameworks exhibiting size-selective catalysis with encapsulated noble-metal nanoparticles. *Adv. Mater.* **26**, 4056–4060 (2014).
20. Wißmann, G. et al. Modulated synthesis of Zr-fumarate MOF. *Microporous Mesoporous Mater.* **152**, 64–70 (2012).
21. Li, Y., Yang, F.H. & Yang, R.T. Kinetics and mechanistic model for hydrogen spillover on bridged metal–organic frameworks. *J. Phys. Chem. C* **111**, 3405–3411 (2007).
22. Tan, K. et al. Reactivity of atomic layer deposition precursors with OH/H₂O-containing metal organic framework materials. *Chem. Mater.* **31**, 2286–2295 (2019).
23. Chen, H. et al. Synergistically assembled MWCNT/graphene foam with highly efficient microwave absorption in both C and X bands. *Carbon* **124**, 506–514 (2017).
24. Hou, S. et al. Modulating in-plane defective density of carbon nanotubes by graphitic carbon nitride quantum dots for enhanced triiodide reduction. *Adv. Funct. Mater.* **2023**, 2212112 (2023).

REVIEWERS' COMMENTS

Reviewer #1 (Remarks to the Author):

As I mentioned in my previous review, the concept of enhancing H spillover with the presence of H₂O is not new. The only novelty of the present study, in comparison to earlier ones, is the use of a different material, MOF, for investigating hydrogen spillover. However, I believe that the experiments in the present study were conducted with sufficient scientific rigor. Additionally, considering the significant growth of the MOF community, the paper has the potential to attract a substantial readership. Therefore, I maintain a neutral stance and hope that the editor can make a final decision by considering the comments of the other two reviewers.

Reviewer #2 (Remarks to the Author):

According to the reviewer's requirements, the authors revised the manuscript. The results of this manuscript are surely interesting and important for deeper understanding of hydrogen spillover in porous crystalline materials and designing more efficient hydrogenation or dehydrogenation composite catalysts. I would like to accept the manuscript for publication in *Advanced Materials*.

Reviewer #3 (Remarks to the Author):

Most of my concerns have been addressed, and now I would like to recommend this paper for publication. However, with regard to my second question, I would like to remind the author to pay attention to the difference between region and diameter and not to mislead the potential readers.

RESPONSE TO REVIEWERS' COMMENTS

Dear reviewers,

Thank you for reviewing our manuscript for publication. Our manuscript has benefited greatly from the insightful comments. And we are delighted to have been acknowledged with reviewers' help to publish on *Nature Communications*.

Reviewer 1:

As I mentioned in my previous review, the concept of enhancing H spillover with the presence of H₂O is not new. The only novelty of the present study, in comparison to earlier ones, is the use of a different material, MOF, for investigating hydrogen spillover. However, I believe that the experiments in the present study were conducted with sufficient scientific rigor. Additionally, considering the significant growth of the MOF community, the paper has the potential to attract a substantial readership. Therefore, I maintain a neutral stance and hope that the editor can make a final decision by considering the comments of the other two reviewers.

Reply: Thank you very much for the constructive comments. As you have mentioned, MOFs have been widely used in various fields due to their unique structures and performances. Especially in the field of catalysis, MOFs have gained interest from both scientific community and industry as catalytic substrates. Therefore, we anticipate that our exploration of hydrogen spillover in MOFs is constructive and will open up new avenues for investigation into the disciplines of MOF catalysis and hydrogen storage. Finally, thank you again for your sincere comments, which encourage us to think more carefully and deeply about our work.

Reviewer 2:

According to the reviewer's requirements, the authors revised the manuscript. The results of this manuscript are surely interesting and important for deeper understanding of hydrogen spillover in porous crystalline materials and designing more efficient hydrogenation or dehydrogenation composite catalysts. I would like to accept the manuscript for publication in *Advanced Materials*.

Reply: We sincerely appreciate the reviewer's positive comments and kind recognition. And we anticipate these studies will advance the design and application of MOF-based catalysts and the investigation of the effect of hydrogen spillover. Thank you for your support and valuable advice.

Reviewer 3:

Most of my concerns have been addressed, and now I would like to recommend this paper for publication. However, with regard to my second question, I would like to remind the author to pay attention to the difference between region and diameter and not to mislead the potential readers.

Reply: We thank the reviewer very much for the recognition of our work. For a better understanding, additional discussion concerning spillover region and spillover distance has been provided. The reminder on the difference between spillover region and spillover distance has been added in Supporting Discussion in Page 15.

“Particularly, the nearly 100% ligand conversion (Supplementary Fig. 23) and the obvious adhesion phenomenon (Supplementary Fig. 24b) of Pt@150 nm MOF-801 under high pressure suggested a larger spillover region, which essentially spanned around 150 nm in diameter of the whole MOF particles. It's important to take notice that the spillover region, which has a diameter of 100 or 150 nm, refers to the coverage of hydrogen spillover rather than the spillover distance¹. Due to the accumulation of Pt nanoparticles within a central range of approximately 50 nm in diameter in both Pt@110 nm MOF-801 and Pt@150 nm MOF-801, the spillover regions may be converted to spillover distances of at least 30 nm at atmospheric pressure and 50 nm at higher humidity and pressure, which is based on the distance from the accumulation edge of the Pt nanoparticles to the MOF surface.”

Thank you so much for your kind consideration, and we are looking forward to your response.

Sincerely yours,

Fengwei Huo

References:

1. Karim, W. et al. Catalyst support effects on hydrogen spillover. *Nature* **541**, 68–71 (2017).